# IRBIT controls apoptosis by interacting with the Bcl-2 homolog, Bcl2l10, and by promoting ER-mitochondria contact

Benjamin Bonneau[1]*, Hideaki Ando[1], Katsuhiro Kawaai[1], Matsumi Hirose[1], Hiromi Takahashi-Iwanaga[2], Katsuhiko Mikoshiba[1]*

[1]Laboratory for Developmental Neurobiology, RIKEN Brain Science institute, Wako-shi, Japan; [2]Department of Anatomy, School of Medicine Hokkaido University, Sapporo, Japan

**Abstract** IRBIT is a molecule that interacts with the inositol 1,4,5-trisphosphate ($IP_3$)-binding pocket of the $IP_3$ receptor ($IP_3R$), whereas the antiapoptotic protein, Bcl2l10, binds to another part of the $IP_3$-binding domain. Here we show that Bcl2l10 and IRBIT interact and exert an additive inhibition of $IP_3R$ in the physiological state. Moreover, we found that these proteins associate in a complex in mitochondria-associated membranes (MAMs) and that their interplay is involved in apoptosis regulation. MAMs are a hotspot for $Ca^{2+}$ transfer between endoplasmic reticulum (ER) and mitochondria, and massive $Ca^{2+}$ release through $IP_3R$ in mitochondria induces cell death. We found that upon apoptotic stress, IRBIT is dephosphorylated, becoming an inhibitor of Bcl2l10. Moreover, IRBIT promotes ER mitochondria contact. Our results suggest that by inhibiting Bcl2l10 activity and promoting contact between ER and mitochondria, IRBIT facilitates massive $Ca^{2+}$ transfer to mitochondria and promotes apoptosis. This work then describes IRBIT as a new regulator of cell death.

*For correspondence: benjamin. bonneau@riken.jp (BB); mikosiba@ brain.riken.jp (KM)

**Competing interests:** The authors declare that no competing interests exist.

## Introduction

Elevation of intracellular $Ca^{2+}$ concentration serves as a second messenger for numerous processes, including the cell cycle, fertilization or apoptosis (*Berridge et al., 2003*). At the endoplasmic reticulum (ER), $Ca^{2+}$ signals are mainly generated by the $Ca^{2+}$ channel inositol-1,4,5-trisphosphate receptor ($IP_3R$) in response to $IP_3$ binding. The cellular response to a $Ca^{2+}$ signal depends on the amplitude and frequency of this signal. $IP_3R$ is then tightly regulated by post-translational modifications and interacting partners that modulate $Ca^{2+}$ release according to cellular context (*Mikoshiba, 2007*; *Foskett et al., 2007*).

In particular, several Bcl-2 family proteins have been reported to regulate $IP_3R$ activity (*Bonneau et al., 2013*). These proteins are well known for their role in apoptosis through the control of outer mitochondrial membrane permeabilization, cytochrome c release, and subsequent activation of caspases (*Youle and Strasser, 2008*). However, Bcl-2 family proteins are also involved in $Ca^{2+}$-induced apoptosis. The correct functioning of mitochondria requires $Ca^{2+}$, which is supplied by some portions of ER that are in physical contact with the mitochondria (called MAMs for mitochondria-associated ER membranes). In MAMs, $IP_3R$ associates with the mitochondrial voltage-dependent anion channel (VDAC), allowing a direct transfer of $Ca^{2+}$ into the mitochondria (*Szabadkai et al., 2006*). However, if the amount of $Ca^{2+}$ transferred is too high, it induces cytochrome c release and apoptosis. MAMs are then acknowledged to be an essential component of $Ca^{2+}$-induced apoptosis (*Giorgi et al., 2009*). In this regard, some Bcl-2 family proteins localize at ER and modulate $Ca^{2+}$ release (*Bonneau et al., 2013*). Several antiapoptotic members notably interact with $IP_3R$, each one

regulating channel activity by a distinct mechanism. For example, Bcl-2 interacts with the central part of $IP_3R$ and reduces $Ca^{2+}$ release to inhibit proapoptotic $Ca^{2+}$ signals (*Hanson et al., 2008*; *Rong et al., 2009*). By contrast, Bcl-xL interacts with the most C-terminal domain of $IP_3R$ and stimulates pro-survival $Ca^{2+}$ transfer to the mitochondria (*White et al., 2005*).

Recently, Nrz, a Bcl-2 homolog in zebrafish, was shown to interact with the N-terminal $IP_3$-binding domain of $IP_3R$ and to decrease ligand binding on the receptor, thus reducing $Ca^{2+}$ release from ER (*Bonneau et al., 2014*). Only a few regulators of $IP_3R$ interact with the $IP_3$-binding domain ($IP_3BD$) or act by interfering with ligand fixation. Among them, IRBIT acts in a manner similar to Nrz. IRBIT directly competes with $IP_3$ by interacting with the residues of the $IP_3$ binding domain ($IP_3BD$) that are involved in $IP_3$ binding (known as the $IP_3$-binding pocket). This increases the threshold of $IP_3$ required for $IP_3R$ opening and then decreases $Ca^{2+}$ release (*Ando et al., 2006*). In addition to $IP_3$, IRBIT interacts with various partners, such as ion transporters and exchangers including NBCe1-B (*Shirakabe et al., 2006*), NHE3 (*He et al., 2008*), and Slc26a6 (*Park et al., 2013*), the $Cl^-$ channel CFTR (*Yang et al., 2009*), Fip1 (*Kiefer et al., 2009*), the ribonucleotide reductase (RNR) (*Arnaoutov and Dasso, 2014*) and kinases including $CaMKII\alpha$ (*Kawaai et al., 2015*), PIPKI, and $II\alpha$ (*Ando et al., 2015*). All of these interactions involve the N-terminal region of IRBIT, which contains a serine-rich region with several phosphorylation sites (*Ando et al., 2006*; *Devogelaere et al., 2007*). In particular, phosphorylation of Ser68 is required for the subsequent sequential phosphorylation of Ser71, Ser74, and Ser77 by the casein kinase I (*Ando et al., 2006*; *Devogelaere et al., 2007*). This multiple phosphorylation of IRBIT is critical for its interaction with $IP_3R$ and most of its other partners.

Nrz is the zebrafish ortholog of the mammalian antiapoptotic protein Bcl2l10 (also called Nrh, Bcl-B, or Diva/Boo). To date, little is known about Bcl2l10, particularly regarding its effect on $Ca^{2+}$ signaling. In mammals, Bcl2l10 is mainly expressed in the ovary and testis, but also in the lung and the developing nervous system (*Aouacheria et al., 2001*; *Inohara et al., 1998*). Interestingly, IRBIT is also strongly expressed in these organs (*Ando et al., 2003*). Nrz and IRBIT both regulate $IP_3R$ activity by interacting with the $IP_3BD$. However, they do not share a common binding site on the $IP_3BD$ as Nrz does not interact with the residues required for $IP_3$ and IRBIT binding (*Bonneau et al., 2014*). This suggests that Nrz and IRBIT may interact with the $IP_3BD$ at the same time. In the present study, we showed that, like Nrz, Bcl2l10 interacts with the $IP_3BD$ of $IP_3R$. We then analyzed how IRBIT and Bcl2l10 behave towards each other. We found that IRBIT and Bcl2l10 interacted together independently of IRBIT binding to $IP_3R$ and that they cooperated to regulate $IP_3R$ activity. Furthermore, we showed that these two proteins localized in MAMs, where they are part of a protein complex with $IP_3R$ and VDAC. Unexpectedly, we found that IRBIT promoted cell death through two mechanisms. First during apoptosis, IRBIT was dephosphorylated, and unphosphorylated IRBIT appeared to inhibit the antiapoptotic activity of Bcl2l10. Second, IRBIT appeared to promote contact between ER and mitochondria that may facilitate proapoptotic $Ca^{2+}$ transfer. Considered collectively, our results suggested a strong relationship between IRBIT and Bcl2l10 and pointed out, for the first time, the implied involvement of IRBIT in cell death.

## Results

### Bcl2l10 interacts with $IP_3BD$ and reduces $Ca^{2+}$ release from $IP_3R$

The Bcl2l10 orthology group is highly divergent (*Guillemin et al., 2011*) and human Bcl2l10 only shares 28.4% identity with Nrz (*Figure 1A*). We therefore first investigated whether Bcl2l10 behaves like Nrz and interacts with $IP_3R$. The three different isoforms of $IP_3R$ were immunoprecipitated from HeLa cells extract, and we found that endogenous Bcl2l10 interacted with the three isoforms, although the interaction with $IP_3R2$ appeared weaker than that of $IP_3R1$ or $IP_3R3$ (*Figure 1B*). We next evaluated whether Bcl2l10 also interacts with the $IP_3BD$ (amino acids 224–604 of $IP_3R1$) as does Nrz. Extracts of HeLa cells expressing FLAG-Bcl2l10 were subjected to GST-pulldown assay, and we demonstrated that Bcl2l10 interacts with recombinant GST-$IP_3BD$ (*Figure 1C*). Nrz as well as Bcl-2 and Bcl-xL were shown to interact with $IP_3R$ via their N-terminal BH4 domain (*Bonneau et al., 2014*; *Monaco et al., 2012*). The deletion of the BH4 domain of Bcl2l10 ($\Delta$BH4Bcl2l10) suppressed its interaction with GST-$IP_3R\Delta CD$ (a protein that has $IP_3R$ deleted from its channel domain) and GST-$IP_3BD$ (*Figure 1C*), demonstrating that Bcl2l10 interacted with $IP_3R$ via its BH4 domain.

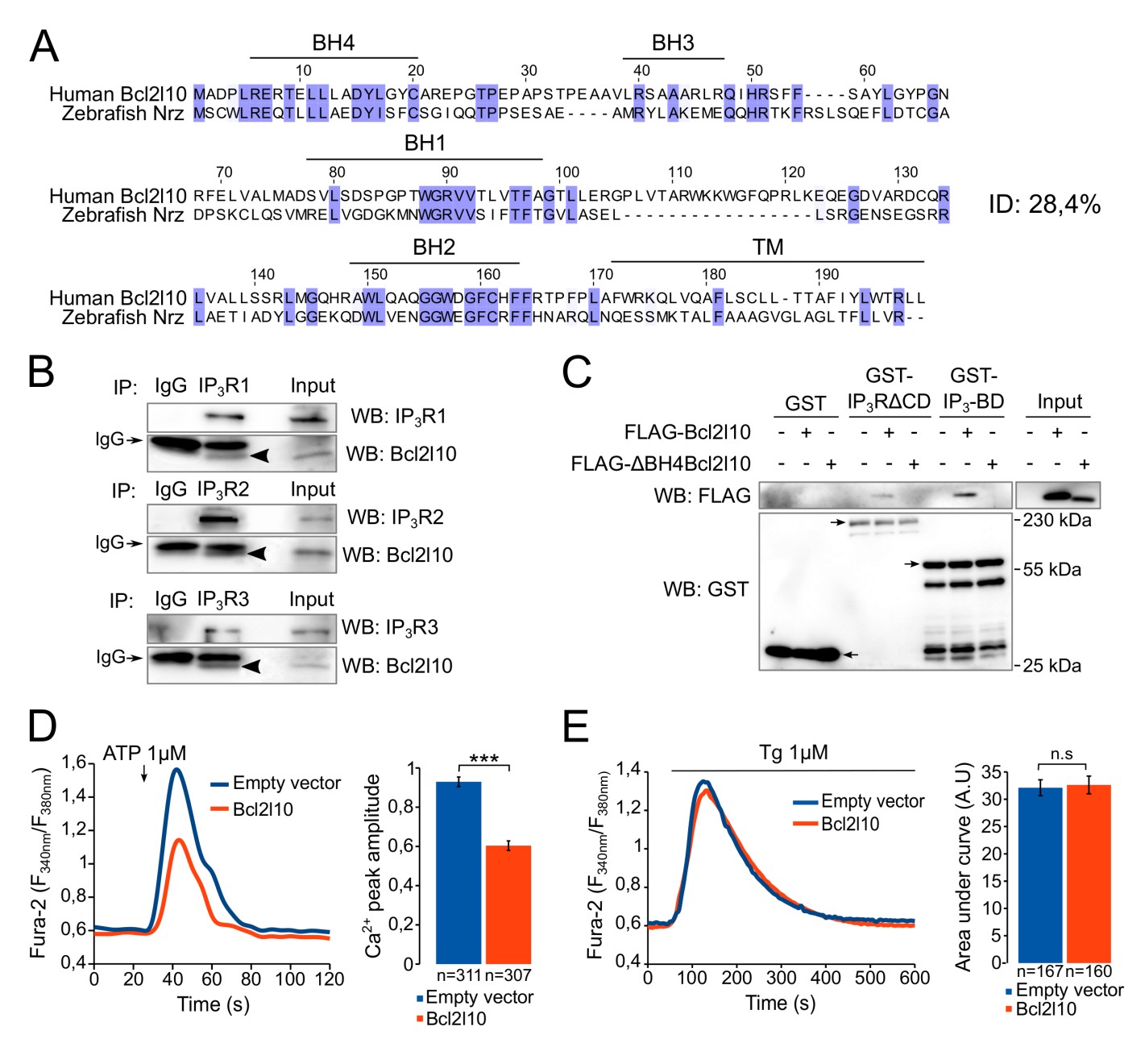

**Figure 1.** Bcl2l10 interacts with and regulates IP3R. (A) Clustal Omega alignment of human Bcl2l10 and zebrafish Nrz primary structures. Identical residues are boxed in blue. The positions of conserved Bcl-2 homology (BH) domains and of the C-terminal transmembrane (TM) domain are indicated. (B) Western blot of immunoprecipitation (IP) between the three endogenous IP3R isoforms (IP3R1, IP3R2 and IP3R3) and endogenous Bcl2l10. IgG antibodies are indicated by arrows. Arrow heads indicate bands corresponding to Bcl2l10 in IP. Western blots are representative of three independent experiments. (C) Western blot of GST-pulldown performed with GST, GST-IP3RΔCD or GST-IP3BD on lysates of HeLa cells expressing FLAG-Bcl2l10 or FLAG-ΔBH4Bcl2l10. Arrows indicate the bands corresponding to GST, GST-IP3RΔCD and GST-IP3BD. Western blot representative of three independents experiments. (D) Left panel: representative Ca$^{2+}$ response curve of Fura-2-loaded cells stimulated with 1 μM ATP at the indicated times. Cells were transfected with empty vector or FLAG-Bcl2l10. Basal Fura-2 $F_{340\,nm}/F_{380\,nm}$ = 0.61 ± 0.01 and 0.59 ± 0.01 for empty vector and Bcl2l10 cells, respectively. Right panel: Bar graph showing the mean amplitude (±SEM) of the ATP-induced Ca$^{2+}$ peak (n: number of cells analyzed from five independent experiments). (E) Left panel: representative response curve of Fura-2-loaded cells treated with 1 μM thapsigargin at the indicated times. Right panel: Bar graph showing the mean area under curve (AUC) (±SEM) of the thapsigargin-induced Ca$^{2+}$ peak (n: number of cells analyzed from three independent experiments). ***p<0.001; n.s p>0.05.

Subsequently, we examined the effect of Bcl2l10 on $IP_3$-induced $Ca^{2+}$ release (IICR) in cultured cells. In these experiments, mouse embryonic fibroblasts (MEF) were used as they exhibit robust IICR in response to ATP stimulation. Expression of Bcl2l10 significantly reduced IICR following the treatment of cells with 1 µM ATP without affecting basal cytosolic $[Ca^{2+}]$, as measured with the cytosolic $Ca^{2+}$-sensitive dye Fura-2 (*Figure 1D*). It has been shown that Bcl-2 family proteins can modify $Ca^{2+}$ release from the ER by acting on the steady-state concentration of $Ca^{2+}$ in the ER (*Pinton and Rizzuto, 2006*). To address a possible effect of Bcl2l10 on ER $Ca^{2+}$ content, MEF cells were incubated with Fura-2 and treated with 1 µM thapsigargin, which induced the depletion of ER $[Ca^{2+}]$. We found that expression of Bcl2l10 did not alter ER $Ca^{2+}$ content (*Figure 1E*). Considered collectively, these results suggest that Bcl2l10 acts like Nrz in zebrafish and reduces $Ca^{2+}$ release from the ER by interacting with the $IP_3BD$ of $IP_3R$.

## IRBIT and Bcl2l10 exert an additive inhibition of $IP_3R$

IRBIT and Bcl2l10 are among the few regulators of $IP_3R$ that interact with the $IP_3BD$. This raises the possibility of cooperation or, alternatively, competition between these two proteins. To address this question, we first analyzed the effect of IRBIT and Bcl2l10 on IICR. Overexpression of IRBIT in cells expressing the endogenous protein has no effect on IICR (*Ando et al., 2006*). Thus, to avoid the contribution of endogenous IRBIT protein, we used MEF cells derived from IRBIT knockout (KO) mice (*Kawaai et al., 2015*). In these cells, expression of IRBIT or Bcl2l10 alone significantly reduced IICR elicited by ATP treatment compared to the response in the control without affecting basal cytosolic $[Ca^{2+}]$. Interestingly, co-expression of IRBIT and Bcl2l10 had a stronger effect on IICR than expression of each protein alone (*Figure 2A and B*). This result suggested that IRBIT and Bcl2l10 cooperate to exert an additive inhibition of $IP_3R$.

To further explore the cooperation between IRBIT and Bcl2l10, we next studied whether each of these two proteins has an effect on the interaction of the other with the $IP_3BD$. Extracts from cells expressing IRBIT alone or in combination with Bcl2l10 were subjected to GST-pulldown with recombinant GST-$IP_3BD$. When expressed with Bcl2l10, IRBIT appeared to interact more with $IP_3BD$ (*Figure 2C*). In the same way, we performed a GST-pulldown assay with GST-$IP_3BD$ and a recombinant Bcl2l10 protein in the presence or absence of recombinant IRBIT produced in Sf9 cells. This production allows the phosphorylation of IRBIT, which is essential for the interaction of IRBIT with the $IP_3BD$ (*Ando et al., 2006*). Similarly, our results showed that in the presence of IRBIT, Bcl2l10 interacted more strongly with the $IP_3BD$ (*Figure 2D*). Thus, IRBIT and Bcl2l10 appeared to strengthen each other's interaction with $IP_3R$.

IRBIT was first characterized as a protein released from $IP_3R$ by $IP_3$ (*Ando et al., 2003*). Indeed, as $IP_3$ and IRBIT share the same binding site, the binding of $IP_3$ on $IP_3R$ occurs to the detriment of the interaction of IRBIT with $IP_3R$. Thus, reduction of the IRBIT interaction with the $IP_3BD$ in the presence of an increasing concentration of $IP_3$ reflected the binding of $IP_3$ on the receptor (*Figure 2E*). However, in the presence of Bcl2l10, we observed that the effect of $IP_3$ on IRBIT's interaction with the $IP_3BD$ was attenuated (*Figure 2E*). This result confirmed the fact that Bcl2l10 strengthened the interaction of IRBIT with $IP_3R$, and suggested that Bcl2l10 and IRBIT associated to interfere with $IP_3$ binding on the receptor and then reduced $Ca^{2+}$ release from the ER.

## IRBIT and Bcl2l10 interact

The above results suggest that IRBIT and Bcl2l10 can form a regulatory complex on $IP_3R$. Consequently, the possibility that these two proteins interact was investigated. Endogenous IRBIT protein was immunoprecipitated from HeLa cells extract, following which we detected an interaction of this protein with endogenous Bcl2l10 (*Figure 3A*). To further characterize this interaction, we then searched for the domain with which Bcl2l10 and IRBIT are involved. The BH4 domain of Bcl-2 family members mediates their interaction with proteins outside of the Bcl-2 family, such as Raf-1 (*Wang et al., 1996*), calcineurin (*Shibasaki et al., 1997*) and VDAC (*Shimizu et al., 2000*). Co-immunoprecipitation between FLAG-tagged full length Bcl2l10 or ΔBH4Bcl2l10 and HA-IRBIT showed that the BH4 domain of Bcl2l10 was essential for its interaction with IRBIT, as the deletion mutant ΔBH4Bcl2l10 lost its ability to bind IRBIT (*Figure 3B*).

The first 104 amino acids of IRBIT are required for its interaction with $IP_3R$ (*Ando et al., 2003*). In particular, phosphorylation of residues Ser71, Ser74, and Ser77 is essential for IRBIT binding on $IP_3R$.

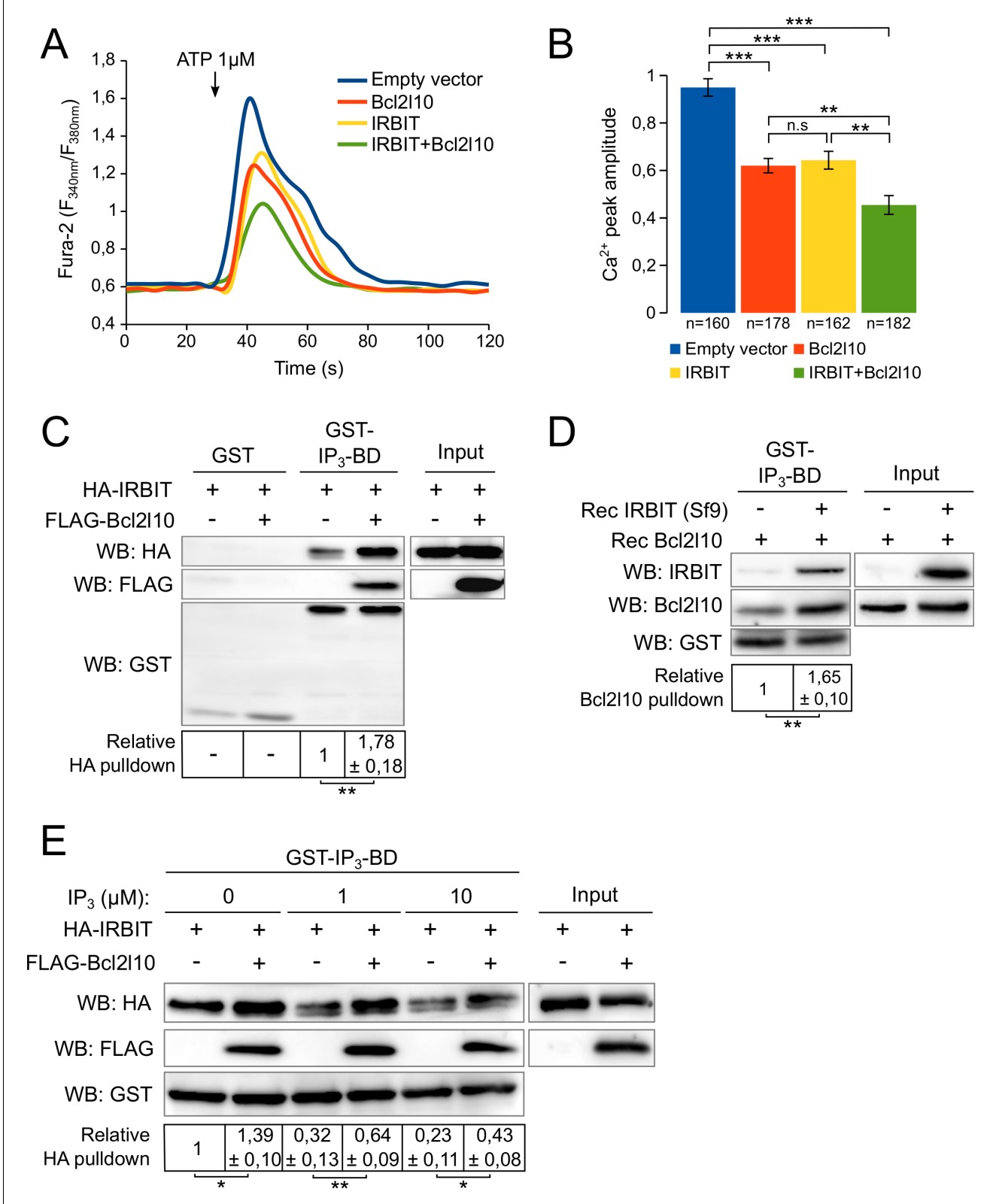

**Figure 2.** IRBIT and Bcl2l10 cooperate to regulate IP$_3$R activity. (**A**) Representative Ca$^{2+}$ response curve of Fura-2-loaded IRBIT KO MEF cells stimulated with 1 μM ATP at the indicated times. Cells were transfected with empty vector or with a plasmid expressing either FLAG-Bcl2l10 or FLAG-IRBIT alone, or FLAG-Bcl2l10 and FLAG-IRBIT together. Basal Fura-2 F$_{340\ nm}$/F$_{380\ nm}$ = 0.61 ± 0.01, 0.59 ± 0.01, 0.59 ± 0.01 and 0.58 ± 0.01 for empty vector, Bcl2l10, IRBIT and Bcl2l10+IRBIT, respectively. (**B**) Bar graph showing the mean amplitude (±SEM) of the ATP-induced Ca$^{2+}$ peak (n: number of cells analyzed

*Figure 2 continued on next page*

*Figure 2 continued*

from five independent experiments). (C) Western blot of GST-pulldown performed with GST or GST-IP$_3$BD on lysates of HeLa cells expressing HA-IRBIT alone or in combination with FLAG-Bcl2l10. Quantification was performed from three independent experiments. (D) Western blot of GST-pulldown performed with GST-IP$_3$BD on recombinant Bcl2l10 alone or in combination with recombinant IRBIT produced in Sf9 cells. Quantification was performed from three independent experiments.(E) Western blot of GST-pulldown performed with GST-IP$_3$BD on lysates of HeLa cells expressing HA-IRBIT alone or in combination with FLAG-Bcl2l10 in the presence of 0, 1 or 10 μM IP$_3$. Quantification was performed from three independent experiments. *p<0.05, **p<0.01, ***p<0.001.

This multiple phosphorylation relies on a prime phosphorylation on Ser68 (*Ando et al., 2006*; *Devogelaere et al., 2007*). Thus, deletion of the first 104 amino acids of IRBIT or the mutation S68A abolishes the interaction with IP$_3$R (*Ando et al., 2003*, *2006*). Interestingly, these mutations did not affect the ability of IRBIT to be co-immunoprecipitated with Bcl2l10 when co-expressed in HeLa cells (*Figure 3D*), suggesting that IRBIT binds Bcl2l10 independently of its interaction with IP$_3$R. To identify the region of IRBIT involved in the interaction with Bcl2l10, we performed additional co-immuno-precipitation experiments between Bcl2l10 and various C-terminus deletion mutants of IRBIT (*Figure 3C*). As expected, the 104 first amino acids of IRBIT (IRBIT 1–104) were not able to interact with Bcl2l10. Similarly, IRBIT 1–138, which contains a coiled-coil domain, did not interact with Bcl2l10 (*Figure 3C and E*). However, we detected a faint interaction between IRBIT 1–169 and Bcl2l10, whereas IRBIT 1–201 and 1–231 interacted strongly with Bcl2l10 (*Figure 3C and E*).

In combination, these results showed that Bcl2l10, through its BH4 domain, interacted with amino acids 138–201 of IRBIT, with residues 169–201 being of particular importance. This portion of IRBIT alone do not bind IP$_3$R, indicating that Bcl2l10–IRBIT interaction did not rely on IRBIT binding to IP$_3$R. This observation is confirmed by the fact that Bcl2l10 could interact with the non-phosphorylat-able mutant IRBIT S68A.

## IRBIT and Bcl2l10 colocalize at ER membranes and belong to the same protein complex in MAM

As IRBIT and Bcl2l10 interact together and with IP$_3$R, we then investigated whether these three proteins can associate to form protein complex in vivo. We first analyzed the subcellular localization of IRBIT and Bcl2l10 to confirm that they colocalize. Subcellular fractionation performed on HeLa cells revealed that IRBIT was present in the cytosol, at the ER, and at the crude mitochondria. This last fraction contains the pure mitochondria and the mitochondria-associated ER membranes (MAMs). IRBIT localized only into MAMs and is not found in pure mitochondria (*Figure 3F*). Bcl2l10 is found at the ER and in the crude mitochondria. However, unlike other antiapoptotic Bcl-2 family members, Bcl2l10 was not detectable in the pure mitochondria fraction and was only present in MAMs (*Figure 3F*). This revealed that Bcl2l10 is an uncommon antiapoptotic protein that exerts its function only at the ER, thereby suggesting that its main mechanism of action may be to regulate Ca$^{2+}$ signaling. In this regard, MAMs play a central role in Ca$^{2+}$-dependent apoptosis because it is in this compartment that Ca$^{2+}$ is directly transferred from the ER to the mitochondria via a protein complex containing IP$_3$R and VDAC (*Szabadkai et al., 2006*). We then investigated whether Bcl2l10 and IRBIT belong to this protein complex, as both proteins localized into MAMs. For this purpose, we analyzed a crude mitochondrial fraction by 2D blue-native SDS-PAGE. By performing a western blot on the second dimension SDS-PAGE, we identified a protein complex that contains IP$_3$R, VDAC, IRBIT, and Bcl2l10 (*Figure 3G*), suggesting that these proteins associate in the native state.

IRBIT and Bcl2l10 then colocalized at the ER membrane, and in particular in MAMs, where they formed a protein complex with IP$_3$R and VDAC. This suggests that IRBIT and Bcl2l10 may be key regulators of Ca$^{2+}$ transfer between ER and mitochondria.

## IRBIT dephosphorylation is involved in the induction of apoptosis

As IRBIT interacts with Bcl2l10, an antiapoptotic protein, and because these proteins associate in a complex in MAMs, a subcellular compartment essential for Ca$^{2+}$-dependent apoptosis, we investigated whether IRBIT plays a role in cell death. To address this, we compared the apoptosis sensitivity of wild type (WT) HeLa cells with IRBIT KO HeLa cells generated using the CRISPR/Cas9 system. Given the localization of IRBIT and Bcl2l10, we focused on Ca$^{2+}$- and ER-stress-induced apoptosis.

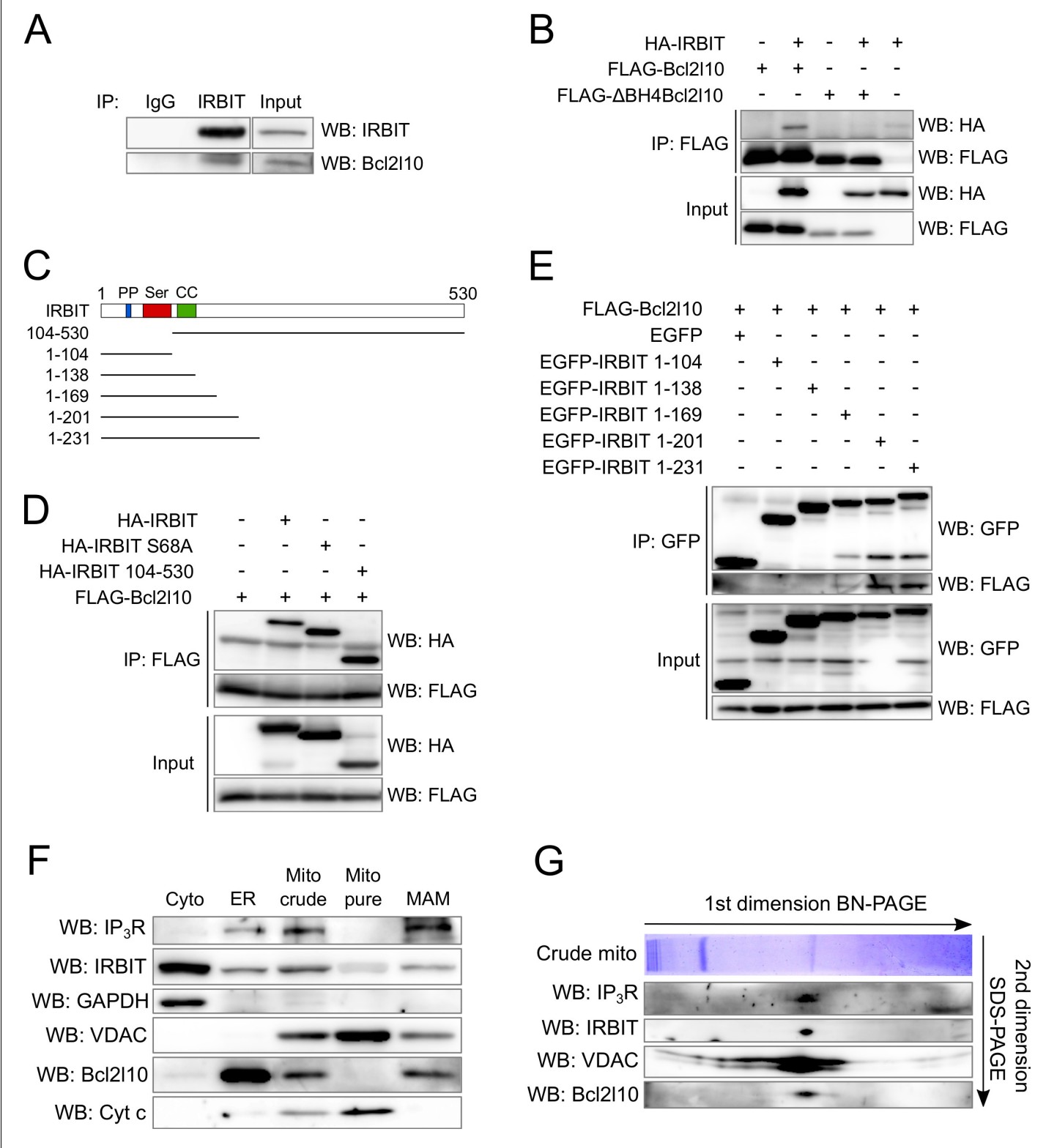

**Figure 3.** IRBIT and Bcl2l10 interact together and belong to a protein complex in MAM. (**A**) Western blot of IP between endogenous IRBIT and endogenous Bcl2l10. Western blot representative of three independent experiments. (**B**) Western blot of IP between HA-IRBIT and FLAG-Bcl2l10 or FLAG-ΔBH4Bcl2l10. Western blot representative of three independent experiments. (**C**) Schematic representation of full-length IRBIT and of the deletion mutants used. PP – Protein phosphatase-1 binding site; Ser –Serine-rich region; CC – Coiled-coil domain. (**D**) Western blot of IP between FLAG-Bcl2l10 and HA-IRBIT or the indicated HA-tagged mutants of IRBIT. Western blot representative of three independent experiments. (**E**) Western

*Figure 3 continued on next page*

*Figure 3 continued*

blot of IP between FLAG-Bcl2l10 and EGFP or the indicated mutants of IRBIT in fusion with EGFP. (**F**) Protein components of subcellular fractions prepared from HeLa cells revealed by Western blot analysis. Cyto – Cytosol; ER – Endoplasmic reticulum; Mito crude – crude mitochondria which is composed of pure mitochondria (Mito Pure) and mitochondria-associated membranes (MAM). Western blot representative of three independent experiments. (**G**) Blue-native (BN) and SDS-PAGE 2D separation of a crude mitochondria fraction prepared from HeLa cells. Second dimension SDS-PAGE was analyzed by Western blot. Western blot representative of two independent experiments.

Cells were then treated with either 1 µM staurosporine (STS), a potent apoptosis inducer that notably triggers mitochondrial $[Ca^{2+}]$ elevation (*Prudent et al., 2015*), or with the ER-stress inducer tunicamycin (TUN at 20 µM) , which also mediates apoptosis by inducing a strong elevation of the mitochondrial $[Ca^{2+}]$ (*Deniaud et al., 2008*). Cell death was assessed by active-Caspase-3 staining, and we surprisingly found that IRBIT KO renders cells more resistant to apoptosis (*Figure 4A* and *Figure 4—figure supplement 1A*). This result was confirmed by Western blot of cleaved-PARP, a classical feature of apoptosis, which showed a reduced cleavage of PARP in both IRBIT KO HeLa cells and IRBIT KO MEF cells (*Figure 4B* and *Figure 4—figure supplement 1B*). Expression of IRBIT in KO cells restored apoptosis sensitivity to a level similar to that estimated in WT cells (*Figure 4B*), suggesting that the effect of IRBIT on cell death observed here is specific, and that IRBIT is required for apoptosis.

The activity of IRBIT depends on its phosphorylation status because its interaction with the majority of its partners relies on the phosphorylation of several serine residues (*Ando et al., 2014*). In particular, IRBIT interaction with $IP_3R$ requires phosphorylation of the residues Ser68, Ser71, Ser74, and Ser77 (*Ando et al., 2006*). As $IP_3R$ plays a central role in apoptosis (*Jayaraman and Marks, 1997*; *Szalai et al., 1999*), we then investigated whether the modulation of IRBIT–$IP_3R$ interaction can account for IRBIT involvement in cell death. To address this question, we analyzed the phosphorylation of IRBIT during apoptosis. The lysate of HeLa cells treated with either 1 µM STS for 6 hr or 20 µM TUN for 24 hr and was analyzed by Western blot using two antibodies specific for phospho-IRBIT (S68p/S71p and S74p/S77p) (*Ando et al., 2009*). The ratio between the intensities of the phospho-IRBIT and IRBIT bands was calculated and compared to the ratio of the control (DMSO treatment) to obtain the relative phosphorylation of S68/S71 and S74/77. Using this approach, we determined that following apoptosis induced by staurosporine or tunicamycin, IRBIT phosphorylation on Ser68, Ser71, Ser74, and S77 was significantly reduced (*Figure 4C and D*). This suggests that IRBIT's function, and notably its effect on $IP_3R$, may be modified during apoptosis. This result also raises the question of whether dephosphorylation of IRBIT is an early event that may participate in the induction of apoptosis or if it is merely a consequence of cell death. We then analyzed IRBIT phosphorylation after a short treatment with STS (0.5, 1 or 1.5 hr) or TUN (4, 6 or 8 hr). At these incubation times, apoptosis was not achieved, as shown by the absence of cleaved-PARP (*Figure 4E*). However, as early as 30 min after STS treatment or 4 hr after TUN treatment, phosphorylation of Ser68, Ser71, Ser74, and Ser77 was already significantly reduced (*Figure 4E and F*), suggesting that dephosphorylation of IRBIT may actually participate in the induction of apoptosis.

Next, we assumed that IRBIT dephosphorylation occurring in apoptosis may reduce its interaction with $IP_3R$, following which the subcellular localization of IRBIT is modified. We analyzed the subcellular localization of IRBIT as well as of Bcl2l10, following apoptosis induction by 1 µM staurosporine for 4 hr or 20 µM tunicamycin for 24 hr. As expected, in cells treated with these drugs, IRBIT localization at the ER and in MAMs was greatly reduced, whereas total IRBIT expression was constant (*Figure 4G*). We could not detect an increased amount of cytosolic IRBIT, probably because IRBIT is mainly a cytosolic protein (*Ando et al., 2015*) and because the quantity of protein displaced from ER membranes is not sufficient to modify the amount of IRBIT in the cytosol markedly. Interestingly, we determined that displacement of IRBIT from ER and MAMs was correlated with a reduction of Bcl2l10 in these compartments, particularly in MAMs, and with an elevation of Bcl2l10 in the cytosol (*Figure 4G*).

As unphosphorylated IRBIT interacts with Bcl2l10 (*Figure 3D*) but not with $IP_3R$, these results suggest that after its dephosphorylation at the onset of apoptosis, unphosphorylated IRBIT could remove Bcl2l10 from ER membranes and displace it to the cytosol. Removal of IRBIT and Bcl2l10

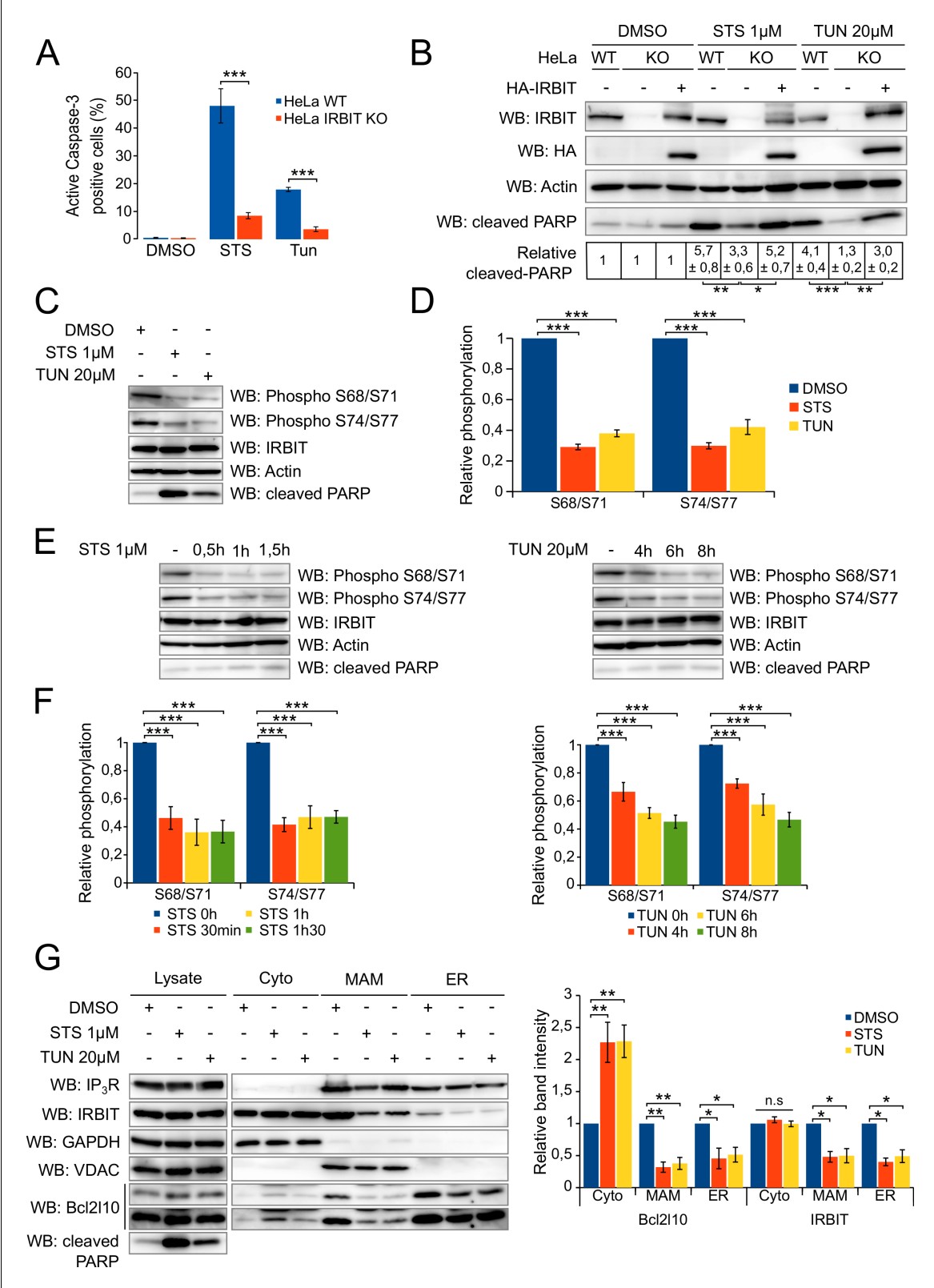

**Figure 4.** IRBIT is required for apoptosis, during which it is dephosphorylated and translocated from ER membranes. (**A**) Bar graph showing the mean percentage (±SEM) of WT or IRBIT KO HeLa cells positive for active-Caspase-3 staining after treatment with DMSO (1/1000 for 24 hr), staurosporine (STS 1 μM for 4 hr) or tunicamycin (Tun 20 μM for 24 hr) (n = three independent experiments, three fields analyzed per condition within each experiment, >200 cells per field). (**B**) Western blot analysis of extracts of WT HeLa cells transfected with empty vector or of IRBIT KO HeLa cells

*Figure 4 continued on next page*

*Figure 4 continued*

transfected with empty vector or HA-IRBIT and treated with DMSO (1/1000 for 24 hr), staurosporine (STS 1 μM for 4 hr) or tunicamycin (Tun 20 μM for 24 hr). Quantification was performed from three independents experiments. (C) Western blot of extracts of HeLa cells treated with DMSO (1/1000 for 24 hr), staurosporine (STS 1 μM for 4 hr) or tunicamycin (Tun 20 μM for 24 hr). (D) Bar graph showing the relative phosphorylation of the Ser68/71 and Ser74/77 residues of IRBIT after staurosporine (1 μM) or tunicamycin (20 μM) treatment for 4 hr or 24 hr, respectively. (n = three independent experiments.) (E) Western blot of extracts of HeLa cells treated with DMSO for 2 hr (−) or with 1 μM STS for 30 min, 60 min or 90 min (left panel) or with DMSO for 8 hr (−) or 20 μM tunicamycin for 4, 6 or 8 hr (right panel). (F) Bar graph showing relative phosphorylation of the Ser68/71 and Ser74/77 residues of IRBIT after STS (1 μM) treatment for 30 min, 60 min or 90 min (left panel) or TUN (20 μM) treatment for 4, 6 or 8 hr (right panel). (n = three independent experiments.) (G) Protein components of subcellular fractions prepared from HeLa cells treated with STS for 4 hr or TUN for 24 hr and revealed by Western blot analysis. Cyto – Cytosol; MAM – mitochondria-associated membranes; ER – Endoplasmic reticulum. *p<0.05, **p<0.01, ***p<0.001. See also *Figure 4—figure supplement 1*.

The following figure supplement is available for figure 4:

**Figure supplement 1.** IRBIT KO protects cells from apoptosis.

from MAMs could consequently facilitate proapoptotic $Ca^{2+}$ transfer to the mitochondria, thereby explaining how IRBIT could promote cell death.

## IRBIT KO abolishes the effect of apoptosis-inducing drugs on $Ca^{2+}$ signaling

To validate this hypothesis, we investigated whether apoptotic stresses does indeed promote $Ca^{2+}$ transfer from ER to mitochondria and whether IRBIT is involved in this process. First, HeLa cells were treated, for a short period, with either 1 μM STS (30 min) or 20 μM TUN (4 hr) and their IICR (induced by 1 μM ATP) was subsequently analyzed. At this early time point, IRBIT is dephosphorylated but apoptosis has not yet occurred (*Figure 4E*). Consistent with a previous study (*Li et al., 2009*), STS and TUN treatment of WT cells significantly increased the amount of $Ca^{2+}$ released through $IP_3R$ (*Figure 5A and B*). STS treatment also slightly increased basal cytosolic $Ca^{2+}$ concentration, but this cannot account for the drug's effect on IICR. As expected, in the control condition (DMSO), IRBIT KO cells had increased IICR compared to WT cells. However, in IRBIT KO cells, the effect of STS and TUN on the release of $Ca^{2+}$ from the ER was greatly attenuated; in these cells STS had only a slight effect on IICR whereas TUN did not significantly modify it (*Figure 5A–C*). These results show that apoptosis-inducing stresses increase the release of $Ca^{2+}$ through $IP_3R$, and that IRBIT plays a key role in this process.

We then examined if this stress-related increased release of $Ca^{2+}$ from the ER is correlated to an elevation of the mitochondrial $[Ca^{2+}]$, a phenomenon known to trigger apoptosis (*Giorgi et al., 2009*). For this purpose, HeLa cells were loaded with the mitochondrial $Ca^{2+}$ dye Rhod-2 and treated with either 1 μM STS for 90 min or 20 μM TUN for 8 hr. As shown in *Figure 5D and E*, STS and TUN treatment induced, in HeLa WT cells, a robust elevation of the mitochondrial $[Ca^{2+}]$ that was significantly reduced in IRBIT KO cells. This suggests that, following STS and TUN treatment, increased $Ca^{2+}$ release from the ER may be transferred to the mitochondria, which will lead to apoptosis. IRBIT seems essential for this to happen as IRBIT KO prevents both increased $Ca^{2+}$ release and elevation of mitochondrial $[Ca^{2+}]$.

All together, these results support the idea that IRBIT participates in cell death by promoting $Ca^{2+}$ transfer from the ER to the mitochondria. IRBIT dephosphorylation at the onset of apoptosis may account for this function of IRBIT as it could lead to the removal of Bcl2l10 from ER membranes.

## Unphsophorylated IRBIT inhibits Bcl2l10 function at the ER

To confirm this model, we then studied the impact of IRBIT dephosphorylation on Bcl2l10 by investigating the effect of unphosphorylated IRBIT on Bcl2l10 function. We first examined the interaction of Bcl2l10 with $IP_3R$ in the presence or absence of unphosphorylated IRBIT. For this purpose, extracts of HeLa cells expressing Bcl2l10 and IRBIT S68A alone or in combination were subjected to GST-pulldown with recombinant GST-$IP_3R\Delta CD$. As expected, IRBIT S68A did not interact with $IP_3R$ whereas Bcl2l10 did. However, coexpression of IRBIT S68A markedly reduced the interaction of Bcl2l10 with $IP_3R$ (*Figure 6A*). To confirm this result, a GST-pulldown assay was performed using

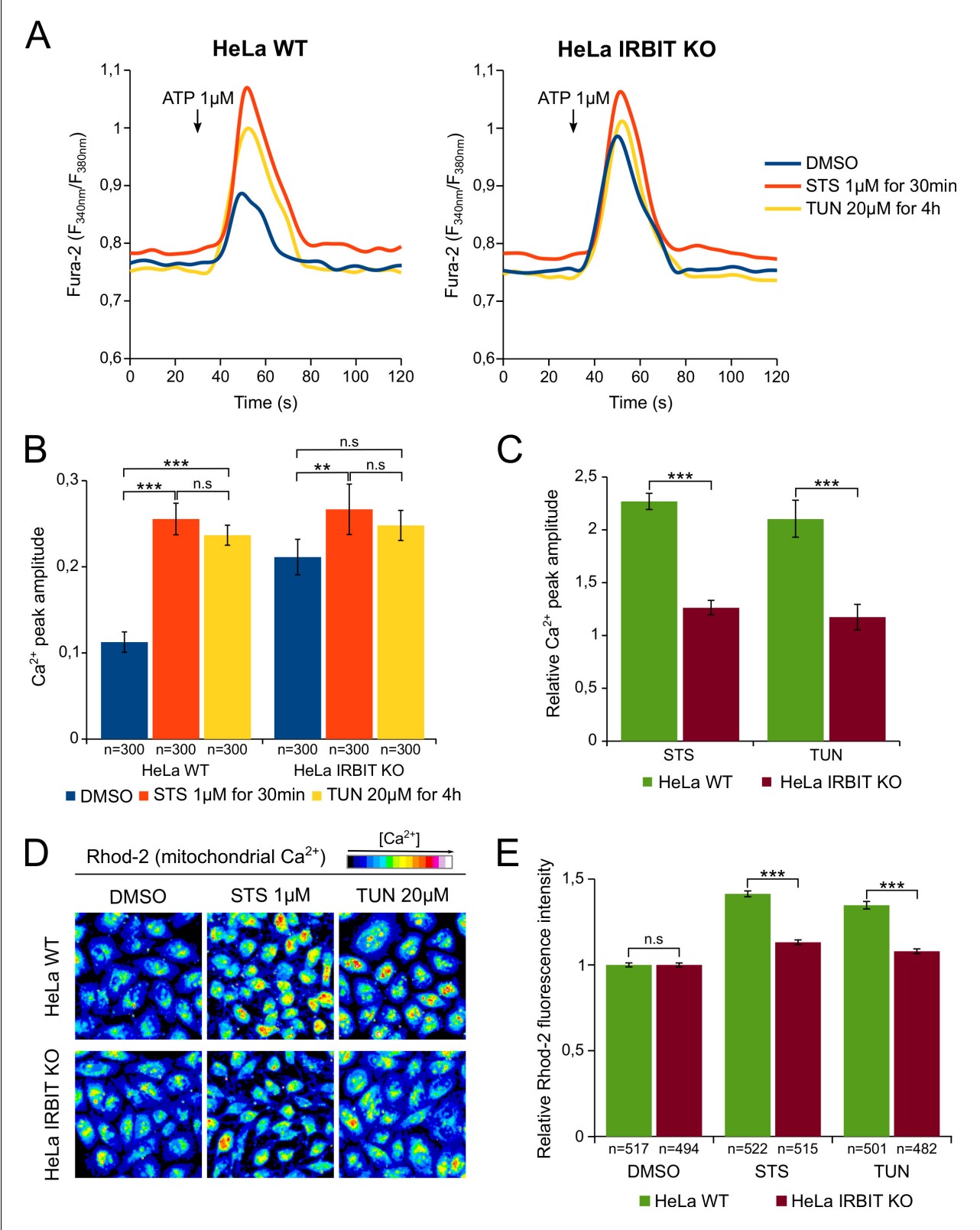

**Figure 5.** IRBIT KO abolishes the effect of apoptosis-inducing drugs on $Ca^{2+}$ signaling. (**A**) Representative $Ca^{2+}$ response curve of Fura-2-loaded HeLa cells stimulated with 1 µM ATP at the indicated times. WT (left panel) or IRBIT KO (right panel) cells were treated with either DMSO (1/1000) for 4 hr, STS 1 µM for 30 min or TUN 20 µM for 4 hr before imaging. Basal Fura-2 $F_{340\,nm}/F_{380\,nm} = 0.76 \pm 0.01$, $0.79 \pm 0.01$ and $0.75 \pm 0.01$ for WT cells treated with DMSO, STS and TUN, respectively; basal Fura-2 $F_{340\,nm}/F_{380\,nm} = 0.75 \pm 0.01$, $0.78 \pm 0.01$ and $0.74 \pm 0.01$ for for IRBIT KO cells treated with DMSO,

*Figure 5 continued on next page*

*Figure 5 continued*

STS and TUN, respectively. (B) Bar graph showing the mean amplitude (±SEM) of the ATP-induced $Ca^{2+}$ peak (n: number of cells analyzed from three independent experiments). (C) Bar graph showing the mean relative amplitude (±SEM) of the ATP-induced $Ca^{2+}$ peak in cells treated with the indicated drugs compared to control (DMSO). (D) Representative images of Rhod-2-loaded HeLa cells treated with either DMSO (1/1000) for 8 hr, STS 1 μM for 90 min or TUN 20 μM for 8 hr. (E) Bar graph showing the mean relative Rhod-2 fluorescence intensity (±SEM) of HeLa cells treated with the indicated drugs compared to control (DMSO). (n: number of cells analyzed from three independent experiments.) *p<0.05, **p<0.01, ***p<0.001.

recombinant GST-IP$_3$BD incubated with recombinant Bcl2l10 in combination with phosphorylated recombinant IRBIT (produced in Sf9 cells) and/or with unphosphorylated recombinant IRBIT (produced in *Escherichia coli*) (*Ando et al., 2006*). In the presence of phosphorylated IRBIT, Bcl2l10 bound to GST-IP$_3$BD strongly, whereas in the presence of unphosphorylated IRBIT, this interaction was significantly weaker (*Figure 6B*). Moreover, although we previously found that phosphorylated IRBIT promotes Bcl2l10's interaction with IP$_3$BD (*Figure 2D*), this effect was abolished in the presence of unphosphorylated IRBIT. Indeed, the interaction between Bcl2l10 and IP$_3$BD in the presence of unphosphorylated IRBIT was similar regardless of whether phosphorylated IRBIT was present (*Figure 6B*). This suggests that unphosphorylated IRBIT inhibits the interaction of Bcl2l10 with IP$_3$R and that dephosphorylation of IRBIT during apoptosis can promote the displacement of Bcl2l10 from ER membranes.

To further corroborate our hypothesis, we assessed the effect of unphosphorylated IRBIT on the ability of Bcl2l10 to decrease $Ca^{2+}$ release through IP$_3$R. As expected, Bcl2l10 alone significantly reduced IICR following treatment with 1 μM ATP, whereas IRBIT S68A alone had no effect on IICR (*Figure 6C and D* and *Figure 6—figure supplement 1A*). However, when these two proteins were coexpressed, we found that Bcl2l10 lost its effect on IICR, as the $Ca^{2+}$ peak was similar to that observed in the absence of Bcl2l10 (*Figure 6C and D*). As IRBIT S68A may displace endogenous Bcl2l10 from ER membranes, the fact that its overexpression alone does not affect IICR was surprising. One reason might be that, in the condition used, the amount of $Ca^{2+}$ released from the ER was high and masked the effect of Bcl2l10 removal from ER membranes. We then overexpressed IRBIT S68A in HeLa IRBIT KO cells , which exhibit weaker $Ca^{2+}$ release following treatment with 1 μM ATP. In these conditions, expression of IRBIT S68A increased IICR to the same extent as Bcl2l10 knockdown, while basal cytosolic $Ca^{2+}$ was unaffected (*Figure 6E,F* and *Figure 6—figure supplement 1B*). Bcl2l10 knockdown using siRNA was confirmed by Western blot (*Figure 6—figure supplement 1B*). These results support the idea that dephosphorylation of IRBIT may have an inhibitory effect on Bcl2l10.

Finally, we examined whether unphosphorylated IRBIT affects the antiapoptotic function of Bcl2l10. HeLa cells expressing Bcl2l10 alone or in combination with IRBIT S68A were treated with 1 μM staurosporine for 6 hr or with 20 μM tunicamycin for 24 hr, and cell death was assessed by staining of active Caspase-3. The overexpression of Bcl2l10 reduced cell death because it is an antiapoptotic protein. However, consistent with our previous results, Bcl2l10 was no longer able to protect cells from apoptosis when coexpressed with IRBIT S68A (*Figure 6G* and *Figure 6—figure supplement 1*), suggesting that unphosphorylated IRBIT acts as an inhibitor of Bcl2l10.

Consequently, all of these results strongly reinforce our model in which dephosphorylation of IRBIT during initiation of apoptosis promotes Bcl2l10 displacement from ER membranes by inhibiting the interaction of Bcl2l10 with IP$_3$R. As a consequence, as suggested by our results, Bcl2l10 may no longer be able to regulate $Ca^{2+}$ release from the ER, and to exert its antiapoptotic activity.

## IRBIT promotes ER-mitochondria $Ca^{2+}$ transfer and the formation of contact points

Our results suggest that, during apoptosis, IRBIT may facilitate the transfer of $Ca^{2+}$ between ER and mitochondria in paticular by inhibiting Bcl2l10. This mechanism should notably occur in MAMs because IRBIT and Bcl2l10 are removed from this compartment, where proapoptotic $Ca^{2+}$ signals aiming at mitochondria are usually generated (*Giorgi et al., 2009*), during apoptosis. To further decipher the role of IRBIT in MAMs, we then examined the extent to which this protein can influence $Ca^{2+}$ flux between ER and mitochondria by comparing ER-mitochondria $Ca^{2+}$ transfer between WT and IRBIT KO MEF cells. In these experiments, cells were treated with 20 μM ATP to induce a

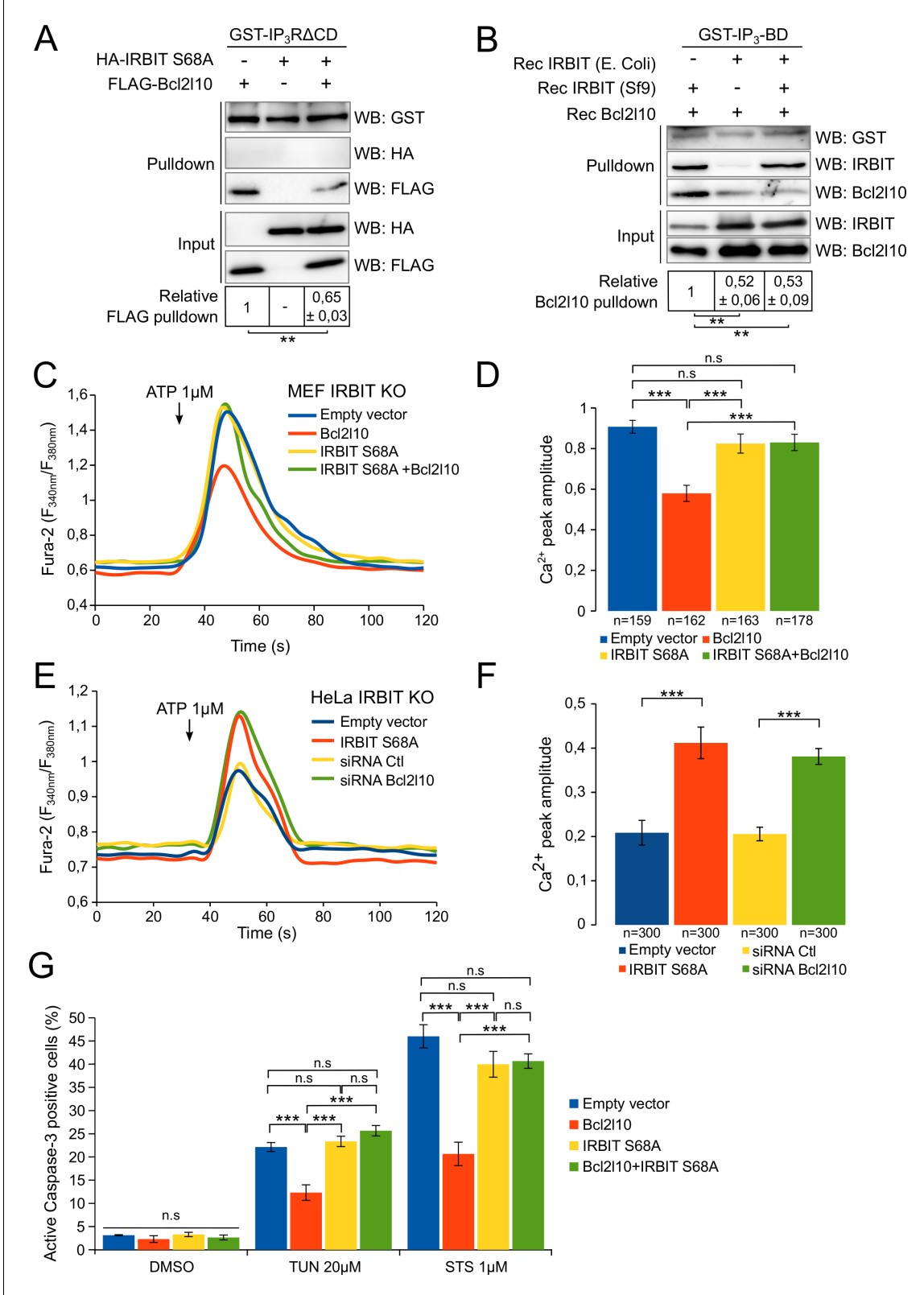

**Figure 6.** Unphosphorylated IRBIT inhibits Bcl2l10 activity. (**A**) Western blot of GST-pulldown performed with GST-IP$_3$RΔCD on lysates of HeLa cells expressing HA-IRBIT S68A and FLAG-Bcl2l10 alone or in combination. Quantification was performed from three independent experiments. (**B**) Western blot of GST-pulldown performed with GST-IP$_3$BD on recombinant Bcl2l10 in combination with recombinant IRBIT produced in Sf9 cells or in *E. coli*. Quantification was performed from three independent experiments. (**C**) Representative Ca$^{2+}$ response curve of Fura-2-loaded IRBIT KO MEF cells

*Figure 6 continued on next page*

*Figure 6 continued*

stimulated with 1 μM ATP at the indicated times. Cells were transfected with empty vector or with a plasmid expressing FLAG-Bcl2l10 and FLAG-IRBIT S68A alone or together. (D) Bar graph showing the mean amplitude (±SEM) of the ATP-induced $Ca^{2+}$ peak (n: number of cells analyzed from three independent experiments). (E) Representative $Ca^{2+}$ response curve of Fura-2-loaded IRBIT KO HeLa cells stimulated with 1 μM ATP at the indicated times. Cells were transfected with empty vector or with a plasmid expressing FLAG-IRBIT S68A or with a control siRNA or with a siRNA against Bcl2l10. (F) Bar graph showing the mean amplitude (±SEM) of the ATP-induced $Ca^{2+}$ peak (n: number of cells analyzed from three independent experiments). (G) Bar graph showing the mean percentage (±SEM) of HeLa cells expressing FLAG-Bcl2l10 and FLAG-IRBIT S68A alone or in combination and positive for active-Caspase-3 staining after treatment with DMSO (1/1000 for 24 hr), tunicamycin (TUN; 20 μM for 24 hr) or staurosporine (STS; 1 μM for 4 hr) (n = 3 independent experiments, three fields analyzed per condition within each experiment, >200 cells per field). *p<0.05, **p<0.01, ***p<0.001. See also *Figure 6—figure supplement 1*.

The following figure supplement is available for figure 6:

**Figure supplement 1.** Unphosphorylated IRBIT inhibits Bcl2l10 anti-apoptotic activity.

massive $Ca^{2+}$ release from the ER, the transfer of which to mitochondria can be easily detected. Under these conditions, as shown by the measurement of cytosolic $Ca^{2+}$ levels with Fura-2, $Ca^{2+}$ release from the ER was greater in IRBIT KO cells than in WT cells (*Figure 7A* and *Figure 7—figure supplement 1A*), which is consistent with the previously described effect of IRBIT on IICR. However, the measurement of mitochondrial $Ca^{2+}$ levels with the mitochondrial $Ca^{2+}$ dye Rhod-2 revealed that although IICR was enhanced in IRBIT KO cells, the amount of $Ca^{2+}$ accumulated in the mitochondria of these cells was lower compared than that in the WT cells (*Figure 7B*). This result shows that $Ca^{2+}$ transfer between ER and mitochondria is impaired in IRBIT KO cells.

ER to mitochondria $Ca^{2+}$ transfer occurs at the level of MAMs (*Giorgi et al., 2009*). We then studied the status of ER-mitochondria contact in IRBIT KO HeLa cells using electron microscopy (*Figure 7C and D*). This analysis revealed that, in IRBIT KO cells, the percentage of mitochondria in contact with ER was less than that in WT cells. Moreover, although the average distance between ER and mitochondria at contact points was similar in the WT and IRBIT KO cells, the average length of contact points was significantly shorter in IRBIT KO cells. Accordingly, in IRBIT KO cells, contact points longer than 600 nm barely existed, whereas those shorter than 200 nm were frequent. By contrast, in WT cells, contact points shorter than 200 nm were rarely found, and those longer than 600 nm were frequent (*Figure 7D*). These results suggest that IRBIT may participate in the formation or stabilization of ER-mitochondria contact points. To further support this observation, we stained the ER and mitochondria of HeLa and MEF cells, and we estimated the colocalization between the two organelles using the Mander's overlap coefficient and the Pearson coefficient (*Bolte and Cordelières, 2006*). As expected, in HeLa IRBIT KO cells and in MEF IRBIT KO cells, colocalization between ER and mitochondria was reduced compared to that in WT cells (*Figure 7E* and *Figure 7—figure supplement 1B*). Expression of IRBIT in IRBIT KO cells increased colocalization coefficients to levels similar to those of WT cells (*Figure 7E*). These results support the idea that IRBIT is involved in the formation or stabilization of ER-mitochondria contact points. As IP$_3$R is a key component of MAMs and as IRBIT interacts with it, we next examined whether this interaction may account for the role of IRBIT in contact points. Interestingly, expression of IRBIT S68A (which cannot bind IP$_3$R) in IRBIT KO cells failed to restore ER-mitochondria contact (*Figure 7E*), suggesting that the interaction between IRBIT and IP$_3$R may be involved in the formation of contact points.

Considered collectively, these results led us to construct the model described below (*Figure 8*). IRBIT promotes ER-mitochondria contact points, thereby facilitating $Ca^{2+}$ transfer to the mitochondria. In the absence of stress, this is counterbalanced by the interaction between IRBIT and Bcl2l10 that controls the amount of $Ca^{2+}$ released through IP$_3$R and allows the correct $Ca^{2+}$ traffic between ER and mitochondria. However, under a stress condition, dephosphorylation of IRBIT induces the displacement of both this protein and Bcl2l10 from ER membranes. This translocation likely favors proapoptotic $Ca^{2+}$ transfer from the ER to the mitochondria, which is facilitated by the close proximity between the organelles promoted by IRBIT. By contrast, in IRBIT KO cells, the lack of IRBIT leads to reduced ER-mitochondria contact; Bcl2l10 may also be more abundant in MAMs and no longer displaced during apoptosis. This may avoid proapoptotic $Ca^{2+}$ transfer between ER and mitochondria, thereby explaining the resistance of these cells to apoptosis.

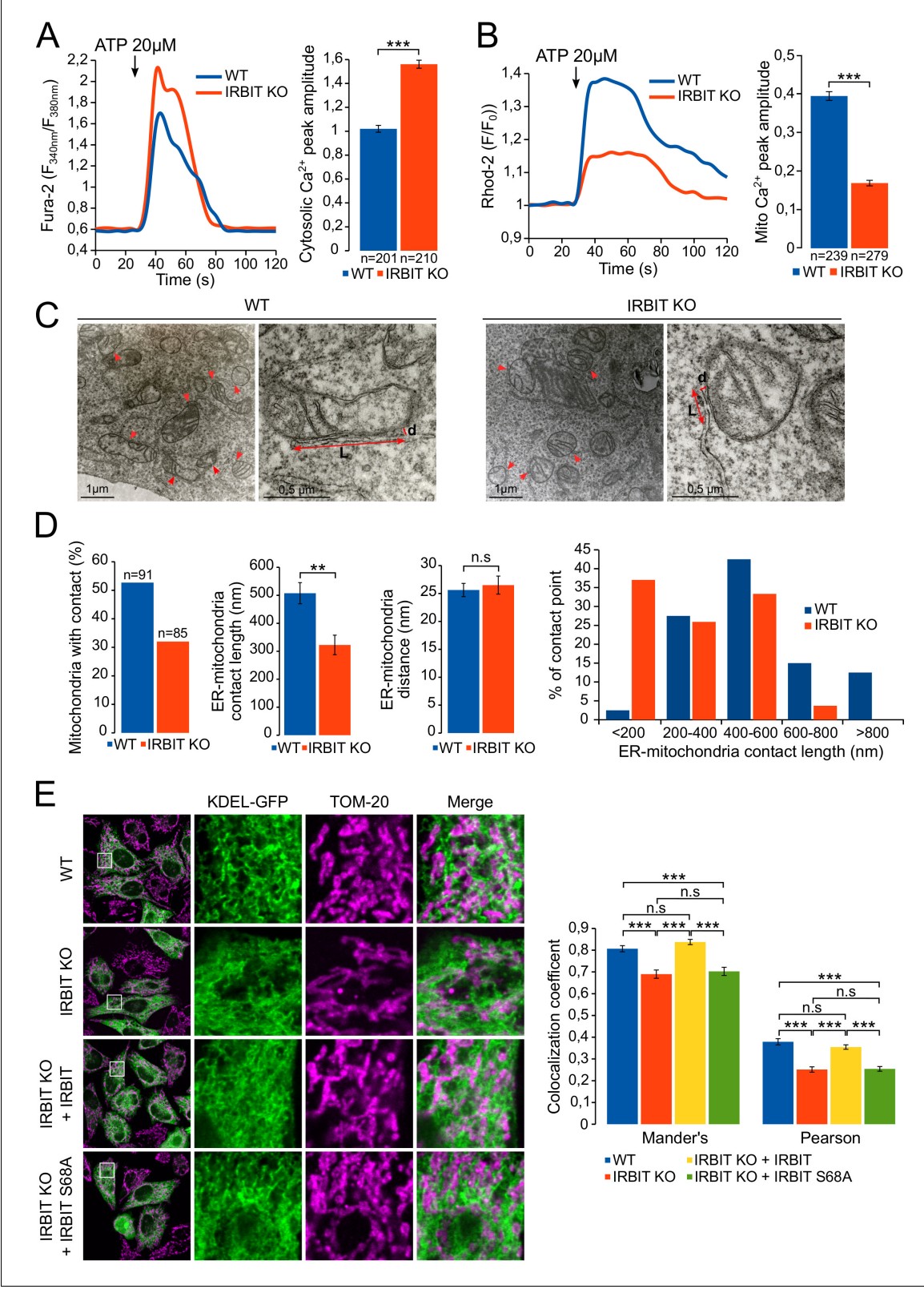

**Figure 7.** IRBIT promotes ER-mitochondria Ca²⁺ transfer and contact. (**A**) Left panel: representative Ca²⁺ response curve of Fura-2 loaded WT or IRBIT KO MEF cells stimulated with 20 µM ATP at the indicated times. Right panel: Bar graph showing the mean amplitude (±SEM) of the ATP-induced Ca²⁺ peak (n: number of cells analyzed from three independent experiments). (**B**) Left panel: representative Ca²⁺ response in the mitochondria of Rhod-2-loaded WT or IRBIT KO MEF cells stimulated with 20 µM at the indicated times. Right panel: bar graph showing the mean amplitude (±SEM) of the

*Figure 7 continued on next page*

*Figure 7 continued*

ATP-induced $Ca^{2+}$ peak in the mitochondria (n: number of cells analyzed from three independent experiments). (**C**) Representative electronic microscopy images of WT and IRBIT KO HeLa cells. Red arrows indicate contact points between ER and mitochondria. Red double-headed arrows indicate the length of the ER-mitochondria contact (L) and the distance between ER and mitochondria (d). (**D**) Quantitative analysis of ER-mitochondria contacts observed by electronic microscopy in WT and IRBIT KO HeLa cells. Bar graphs show the percentage of mitochondria in contact with ER (n = three cells analyzed per condition. WT – 91 mitochondria, IRBIT KO – 85 mitochondria), the mean length (±SEM) of ER–mitochondria contact, the mean distance (±SEM) between ER and mitochondria at contact points and the percentage of ER-mitochondria contact points measuring the indicated length (n = three cells analyzed per condition; WT – 48 mitochondria, IRBIT KO – 27 mitochondria). (**E**) Left panel: immunofluorescence of WT and IRBIT KO HeLa cells transfected with an ER marker (KDEL-GFP (green)) and stained with anti-TOM20 antibody (magenta) for mitochondria labelling. WT cells were also transfected with empty vector and IRBIT KO cells with empty vector or with a vector expressing FLAG-IRBIT or FLAG-IRBIT S68A. Areas shown in close-up highlight ER-mitochondria contact sites. Right panel: Bar graph showing the mean Mander's overlap coefficient (±SEM) and the mean Pearson coefficient (±SEM) of WT and IRBIT KO cells expressing the indicated protein (n = three independent experiments, ~20 cells analyzed per condition for each experiment). *p<0.05, **p<0.01, ***p<0.001. See also *Figure 7—figure supplement 1*.

The following figure supplement is available for figure 7:

**Figure supplement 1.** IRBIT KO reduces ER-mitochondria contact in MEF cells.

## Discussion

In the present study, we demonstrated an interaction between IRBIT and the Bcl-2 homolog, Bcl2l10. These two proteins interact with the same domain of $IP_3R$; but while IRBIT interacts with the $IP_3$-binding pocket (*Ando et al., 2006*), Nrz, the zebrafish ortholog of Bcl2l10, has been shown to interact with different residues (*Bonneau et al., 2014*). Our findings confirm that IRBIT and Bcl2l10 have different binding sites in the $IP_3$-binding domain, and demonstrate the existence of a Bcl2l10-IRBIT complex on $IP_3R$. The BH4 domain of Bcl2l10 is involved in the interaction with IRBIT and $IP_3R$, but two distinct portions of IRBIT mediate the interactions with Bcl2l10 and $IP_3R$. The $IP_3BD$ is structured as a cleft at the end of which the $IP_3$-binding pocket is found (*Bosanac et al., 2002*). The N-terminal part of IRBIT (residues 1–104) is probably buried in the $IP_3BD$ where it interacts with the $IP_3$-binding pocket. We assume that the rest of the protein projects toward the outside of the cleft, where Bcl2l10 is located. This allows residues 169–201 of IRBIT to bind the BH4 domain of Bcl2l10, which is structured in an α-helix; therefore, we can hypothesize that one side of the helix interacts with $IP_3R$ whereas the other interacts with IRBIT. This association between IRBIT and Bcl2l10 may stabilize their interaction with $IP_3R$, thus increasing their effect on IICR.

IRBIT dephosphorylation at the onset of apoptosis appears to participate actively in the execution of cell death. IRBIT possesses a protein phosphatase-1 (PP1) binding site upstream of its serine-rich region (*Figure 3B*), between residues 40 and 44 (*Devogelaere et al., 2007*). This binding site has been shown to dephosphorylate the Ser68 residue, but not residues Ser71, Ser74, and Ser 77 (*Devogelaere et al., 2007*). Interestingly, PP1 was shown to mediate apoptosis via dephosphorylation of Akt (*Thayyullathil et al., 2011*) and pRb (*Puntoni and Villa-Moruzzi, 1999*; *Wang et al., 2001*). Moreover, PP1 can dephosphorylate and activate the BH3-only protein Bad to induce apoptosis (*Ayllón et al., 2000*). Finally, the inhibition of PP1 has been reported to protect cardiomyocytes from tunicamycin-induced apoptosis (*Liu et al., 2014*). All of these studies have highlighted the role of PP1 in the induction of apoptosis, and we can then speculate that IRBIT is a target of PP1 at the onset of apoptosis. However, although dephosphorylation of Ser68 by PP1 may prevent further phosphorylation of Ser71, Ser74, and Ser77, it is unlikely that PP1 by itself accounts for the dephosphorylation of these residues observed during apoptosis. Sequence analysis of IRBIT revealed the existence of a LxVP motif between residues 271 and 274, which is a binding site of the $Ca^{2+}$/calmodulin-dependent phosphatase, calcineurin (*Rodríguez et al., 2009*; *Slupe et al., 2013*). Calcineurin plays a role in apoptosis;notably , it participates in $Ca^{2+}$-dependent apoptosis by activating the BH3-only protein Bad (*Wang et al., 1999*) and by promoting Drp-1 translocation to the mitochondria (*Cereghetti et al., 2008*, *2010*). Further experiments should be performed to study the ability of calcineurin to dephosphorylate IRBIT, but it is likely that the combined action of PP1 and calcineurin is responsible for IRBIT dephosphorylation during apoptosis.

During apoptosis, IRBIT appears to function by displacing Bcl2l10 from ER membranes, thus inhibiting its antiapoptotic function. However, given that IRBIT interacts with numerous other

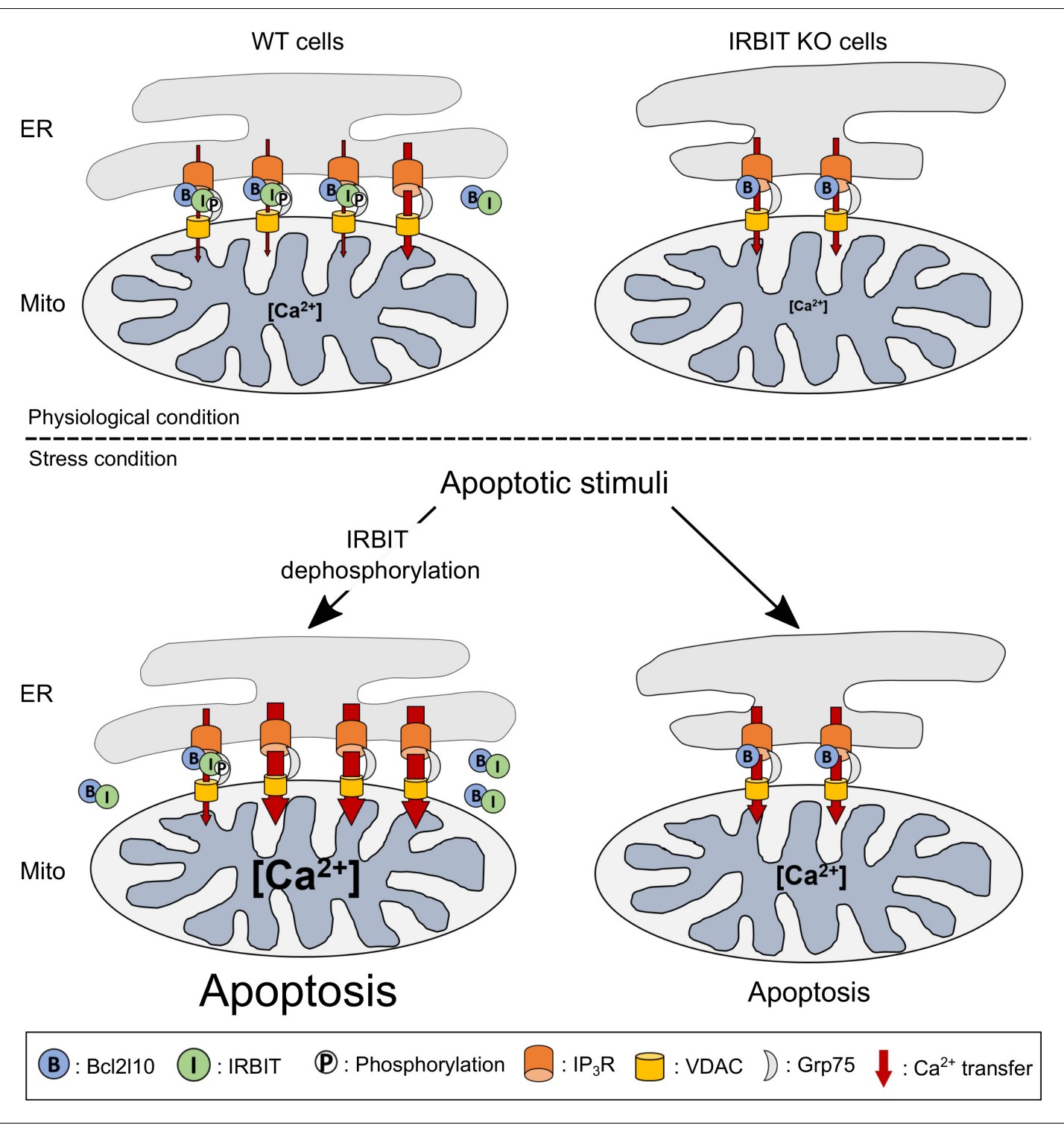

**Figure 8.** Schematic model depicting the role of IRBIT-Bcl2l10 interplay in physiological and stress conditions. In physiological conditions, in WT cells, IRBIT promotes ER-mitochondria contact rendering $Ca^{2+}$ transfer easier between the two organelles. The additive effect of Bcl2l10 and phosphorylated IRBIT on $IP_3R$ maintains $Ca^{2+}$ transfer at a low level. In IRBIT KO cells, the amount of $Ca^{2+}$ that is released through $IP_3R$ is increased due to absence of IRBIT, but $Ca^{2+}$ transfer to the mitochondria is reduced because of the great reduction of ER-mitochondria contact. Following an apoptotic stimuli, $Ca^{2+}$ release from the ER is increased. In WT cells, IRBIT dephosphorylation induces its translocation into the cytosol together with Bcl2l10, allowing a massive $Ca^{2+}$ transfer from ER to mitochondria. By contrast, in IRBIT KO cells, Bcl2l10 is no longer displaced from MAMs, reducing the release of $Ca^{2+}$ from ER. This, combined with the reduction of ER-mitochondria contact, prevents massive $Ca^{2+}$ transfer to mitochondria and thus greatly attenuates apoptosis.

partners, we cannot exclude a contribution of one of these interactions in apoptosis. In particular, the interaction of IRBIT with the $Ca^{2+}$/calmodulin-dependent kinase II$\alpha$ (CamKII$\alpha$) may also play a role in apoptosis. Indeed, IRBIT was shown to inhibit CamKII$\alpha$ (*Kawaai et al., 2015*), which possesses a pro-survival effect, as it notably promotes Bcl-2 expression and inhibits caspase-2 activity (*Nutt et al., 2005*; *Song et al., 2010*; *Wei et al., 2013*). In this regard, inhibition of CamKII was reported to promote apoptosis (*Ma et al., 2009*; *Wei et al., 2015*). Interestingly, as for Bcl2l10, the interaction between CamKII and IRBIT does not depend on IRBIT phosphorylation (*Kawaai et al., 2015*). Thus, in addition to its effect on Bcl2l10, IRBIT dephosphorylation during apoptosis may favor its interaction with CamKII, as it could no longer interact with the majority of its other partners. The increased interaction of IRBIT with CamKII may lead to CamKII inhibition and thus to the promotion of apoptosis.

IRBIT appears to participate in the formation or the stabilization of the contact points between ER and mitochondria. To date, the mechanisms governing the formation of these structures remain largely unknown. Our results suggest that the interaction between IRBIT and IP$_3$R may mediate the effect of IRBIT on ER-mitochondria contact (*Figure 7D*). This result was quite unexpected, as IP$_3$R does not appear to be required for ER-mitochondria contact. Indeed, KO of the three IP$_3$R isoforms in DT40 cells was reported to have no effect on the frequency and length of contact points (*Csordás et al., 2006*). Mitofusin 2 (Mfn2), a GTPase involved in mitochondrial fusion, is acknowledged to be the key player in the establishment of ER-mitochondria contact, as its KO has been reported to greatly affect both the formation of MAMs and $Ca^{2+}$ transfer between the two organelles (*de Brito and Scorrano, 2008*). However, this requirement of Mfn2 is now being reconsidered, as two independent studies concluded that Mfn2 KO actually increases contact between ER and mitochondria (*Cosson et al., 2012*; *Filadi et al., 2015*). At the light of this controversy, we cannot completely rule out the possibility that the role of IP$_3$R in the formation of MAMs has been overlooked, in particular when we take into account the fact that IP$_3$R is part, together with Grp75 and VDAC, of a complex which physically links ER and mitochondria (*Szabadkai et al., 2006*). It is highly likely that this complex is the same as that containing IRBIT and Bcl2l10 (*Figure 3G*). Although Grp75 knockdown altered $Ca^{2+}$ transfer between ER and mitochondria, the role of the IP$_3$R-VDAC complex in ER-mitochondria contact has not been clearly investigated (*Szabadkai et al., 2006*). Additional experiments to reassess the role of this complex and of IP$_3$R in the formation of MAMs may facilitate further understanding of how IRBIT is involved in ER-mitochondria contact. An additional possibility is that IRBIT interacts with different partners in MAMs. Indeed, numerous proteins, such as the Fis1-Bap31 complex (*Iwasawa et al., 2011*), PERK (*Verfaillie et al., 2012*), the VAPB-PTPIP51 complex (*Stoica et al., 2014*), and Drp-1 (*Prudent et al., 2015*) have been shown to localize in MAMs and to regulate ER-mitochondria contact. IRBIT could then associate with some of these proteins to form a tethering complex between ER and mitochondria.

The fact that IRBIT S68A failed to promote ER-mitochondria contact (*Figure 7E*) raises the question of the impact of IRBIT dephosphorylation on contact point stability during apoptosis. Indeed, it can be speculated that IRBIT removal from MAM during apoptosis may phenocopy IRBIT KO and reduce ER-mitochondria contact, thereby decreasing $Ca^{2+}$ transfer to the mitochondria instead of increasing it. However several studies have shown that ER-mitochondria contacts increase during apoptosis (*Csordás et al., 2006*; *Verfaillie et al., 2012*; *Prudent et al., 2015*). In particular, Drp-1 is recruited in ER-mitochondria contact points during apoptosis (*Prudent et al., 2015*) and PERK may also be recruited as its KO abolishes the increase of ER-mitochondria contact points during apoptosis (*Verfaillie et al., 2012*). Our hypothesis, therefore, is that prior to apoptosis, phosphorylated IRBIT promotes ER-mitochondria contact and thus sets up a platform for proper apoptosis. Following the induction of apoptosis, IRBIT is dephosphorylated and translocated into the cytosol, but other proteins such as Drp-1 or PERK are also recruited to MAMs to strengthen ER-mitochondria contact and then compensate IRBIT removal. In IRBIT KO cells, these proteins are probably still recruited, but as ER-mitochondria contacts are smaller, increasing these contacts is not sufficient to insure pro-apoptotic $Ca^{2+}$ transfer.

Our present study revealed a new function of IRBIT as a regulator of cell death through its ability to regulate ER-mitochondria contact and Bcl2l10 activity. Interestingly, IRBIT expression was found to be reduced in a human ovarian cancer cell line (*Jeong et al., 2012*). As Bcl2l10 is highly expressed in the ovary (*Guillemin et al., 2009*; *Inohara et al., 1998*), we can assume that the reduction of IRBIT expression in the ovarian cancer cell line leads to an increased activity of Bcl2l10, which may

contribute to the usual resistance of cancer cells to apoptosis. Moreover, it was recently shown that IRBIT inhibits ribonucleotide reductase (RNR), an enzyme that provides the dNTP pool for DNA replication (*Arnaoutov and Dasso, 2014*). In cancer cells, high activity of RNR is necessary to supply the dNTPs required for rapid cell proliferation, and increased RNR activity is frequently observed in cancer (*Aye et al., 2015*). Thus, reduced expression of IRBIT may contribute to resistance to apoptosis due to increased Bcl2l10 activity and reduced ER-mitochondria contact, and additionally to cell proliferation by promoting increased RNR activity. These observations, associated with the fact that IRBIT expression was found to be reduced in cancer cell lines that are resistant to DNA-damging drugs (*Wittig et al., 2002*), highlight the key role of IRBIT in tumorigenesis, and may lead to a new definition of IRBIT as a tumor suppressor.

## Materials and methods

### Plasmid construction

The sequences coding for Bcl2l10 and ∆BH4Bcl2l10 were amplified from pSG5-FLAG-Bcl2l10, a gift from G. Gillet (*Aouacheria et al., 2001*). They were cloned between BamHI and EcoRI restriction sites of pCDNA3 vector (Invitrogen) containing the coding sequence of FLAG-tag between HindIII and BamHI restriction sites. The sequence coding for Bcl2l10∆TM was amplified from pCDNA3-FLAG-Bcl2l10 and cloned between NdeI and HindIII restriction sites of the bacterial expression vector pET-23a(+) (Novagen).

Vector coding for KDEL-GFP has been described previously (*Bannai et al., 2004*). Expression vector coding HA-IRBIT, HA-IRBIT S68A have been described previously (*Ando et al., 2006*). Deletion mutants of IRBIT were generated by subcloning truncated cDNA fragments of IRBIT between the HindIII and KpnI restriction sites of pHM6 vector (Boehringer Mannheim) or between the HindIII and BamHI restriction sites of pEGFPC1 vector (Clontech). The bidirectional vector pBI-CMV1 (Clontech) containing two MCS under the control of two distinct CMV promoters was modified to give pBI-CMEF by replacing the second CMV promoter by the EF1α promoter. The sequences coding for FLAG-Bcl2l10, FLAG-IRBIT and FLAG-IRBIT S68A were amplified from pCDNA3-FLAG-Bcl2l10, pCDNA3-FLAG-IRBIT and pCDNA3-FLAG-IRBIT S68A (*Ando et al., 2006*) and cloned in pBI-CMEF. FLAG-Bcl2l10 was cloned under the control of EF1α promoter in MCS2 between EcoRI and XbaI restriction sites. FLAG-IRBIT and FLAG-IRBIT S68A were cloned under the control of the CMV promoter in MCS1 between ClaI and EcoRV restriction sites.

### Antibodies

Rabbit antiphospho-IRBIT Ser68p/Ser71p and rabbit antiphospho-IRBIT Ser74p/Ser77p Ab (*Ando et al., 2009*), rabbit anti-IP$_3$Rs (KM1112 for IP$_3$R1, KM1083 for IP$_3$R2 and KM1082 for IP$_3$R3) (*Kawaai et al., 2009*) and rabbit anti-IRBIT antibody (*Ando et al., 2003*) have been described previously. The following antibodies were used: rat anti-HA Ab (3F10, Roche, RRID:AB_390919), mouse anti-β-actin Ab (AC-15, Sigma, RRID:AB_476744), mouse anti-Flag Ab (M2, F3165, Sigma, RRID:AB_259529), rabbit anti-Flag Ab (PA1-984B, Thermofisher, RRID:AB_347227), mouse anti-GFP Ab (B-2, Santa Cruz Biotechnology Inc, RRID:AB_627695), rabbit anti-cytochrome c Ab (H-104, sc-7159, Santa Cruz Biotechnology Inc, RRID:AB_2090474), rabbit anti-Tom20 Ab (FL-145, sc-11415, Santa Cruz Biotechnology Inc, RRID:AB_2207533), mouse anti-AHCYL1/2 Ab (D-7, sc-271581, Santa Cruz Biotechnology Inc, RRID:AB_10649944), mouse anti-VDAC1/Porin Ab (20B12AF2, ab14734, Abcam, RRID:AB_443084), rabbit anti-Bcl2l10 Ab (3869S, Cell Signaling Technology, RRID:AB_2274786), rabbit anti-Cleaved PARP Ab (E51, ab32064, Abcam, RRID:AB_777102), rabbit anti-Cleaved-Caspase-3 Ab (9661S, Cell Signaling Technology, RRID:AB_2341188), mouse anti-GST Ab (B-14, sc-138, Santa Cruz Biotechnology Inc, RRID:AB_627677), and mouse anti-GAPDH Ab (G-9, sc-365062, Santa Cruz Biotechnology Inc, RRID:AB_10847862).

### Cell culture and transfection

Wild-type (WT) or IRBIT knockout (KO) mouse embryonic fibroblast (MEF) cells have been described previously (*Kawaai et al., 2015*). HeLa cells (RRID:CVCL_0030) were obtained from the RIKEN Bio-Resource Center (Ibaraki, Japan) where their identity was checked by Short Tandem Repeat (STR) polymorphism profiling analysis. MEF cells and HeLa cells were cultured at 37°C with 5% $CO_2$ in

Dulbecco's modified essential medium (DMEM, Nacalai Tesque) supplemented with 10% (vol/vol) FBS, 50 units/mL penicillin, and 0.05 mg/mL streptomycin (Nacalai Tesque). None of the used cell lines were in the list of commonly misidentified cell lines maintained by the International Cell Line Authentication Committee. All cell lines were checked for mycoplasma contamination.

HeLa cells were transfected, according to manufacturer's instructions, with X-treamGENE HP reagent (Roche Diagnostics) for plasmids and with Lipofectamine 2000 (Thermo Fisher Scientific) for siRNA. For MEF cells, electroporation was performed using MEF 1 Nucleofector Kit (VPD-1004, Lonza) according to manufacturer's instructions.

## CRISPR-mediated gene targeting

The guide RNA (gRNA) sequences for human IRBIT exon 2 (forward: 5'-CACCGCAAAGATC TTCGGCCAGTTT-3', reverse: 5'-AAACAAACTGGCCGAAGATCTTTGC-3') were ligated into the BbsI sites of pSpCas9(BB)-2A-GFP (PX458) (a gift from Dr. Feng Zhang, Addgene plasmid # 48138) as described previously (Ran et al., 2013). PX458-gRNAs were transfected into HeLa cells using TransIT-LT1 reagent (Mirus) according to manufactures' instructions. Clonal cell lines were isolated by culturing single cells in 96-well plates and were screened by western blotting with anti-IRBIT antibody. Genomic DNA containing gRNA target sites were amplified by PCR using primers flanking IRBIT exon 2 (5'-AAAGGATCCGAACCATGTGATTACATGGC-3', 5'-GGGAAGCTTCAAAGG TGGGCAGTCATAAC-3', restriction enzyme sites are underlined). PCR products were cloned into BamHI-HindIII sites of pBluescript II (Stratagene) and mutations were confirmed by DNA sequencing.

## Recombinant protein

Recombinant GST, GST-IP$_3$R$\Delta$CD (also described as GST-EL), GST-IP$_3$BD production (Ando et al., 2003) and recombinant IRBIT production expressed in *E. coli* or Sf9 cells have been described previously (Ando et al., 2006).

For recombinant Bcl2l10-His production, pET-23a(+)-Bcl2l10$\Delta$TM was transformed in BL-21. Bacteria were grown at 37°C in LB medium containing Ampicilline (50 µg/mL) to a cell density of 0.7–0.8 (600 nm) then 0.5 mM IPTG (isopropyl β-D-thiogalactoside) was added and the cultures were incubated at 25°C overnight before collection of cells by centrifugation at 6000 x *g* for 10 min. The pellet was resuspended in purification buffer (50 mM Na$_2$HPO$_4$, 500 mM Nacl, pH 8), lysed by sonication for 5 min and the mixture was centrifuged at 15,000 x *g* for 30 min. The resulting supernatant was incubated for 30 min at room temperature with ProBond column (Life technologies) which was pre-incubated in binding buffer (8 M Urea, 20 mM Na$_2$HPO$_4$,500 mM Nacl, pH 7.8). Resin was washed twice with binding buffer then twice with binding buffer pH6 and finally four times with purification buffer containing 50 mM Imidazole. Elution was performed with purification buffer containing 250 mM imidazole and resulting sample was dialysed with 50 mM TrisHcl pH 8, 1 mM EDTA, 1 mM β-mercatpoethanol and then concentrated using Vivaspin6 (GE lifesciences).

## Immunoprecipitation and GST-pulldown

For immunoprecipitation, HeLa cells were washed with PBS and then solubilized for 30 min at 4°C in TNE buffer (10 mM Tris-HCl, pH 7.4, 200 mM NaCl, 1 mM EDTA, 0,2% NP-40) with proteinase inhibitor (complete, Roche). The lysate was centrifuged at 4°C for 20 min at 16,000 x *g*. The supernatant was pre-cleared with Protein-G sepharose 4B Fastflow (Protein-G, GE Healthcare) and then incubated at 4°C for 4 hr with the indicated antibody and Protein-G sepharose 4B Fastflow. Beads were washed three times with TNE buffer. Precipitated proteins were eluted by boiling in SDS-PAGE sample buffer and analyzed by immunoblotting with appropriate antibodies.

For GST pull-down assay, cell lysates prepared as described above were incubated with 10 µg GST fusion proteins for 1 hr at 4°C. After the addition of glutathione-Sepharose 4B (GE Healthcare), the samples were incubated for 1 hr at 4°C. The resins were washed three times with TNE buffer, and bound proteins were eluted with 20 mM glutathione, mixed with SDS-PAGE sample buffer and analyzed by immunoblotting with appropriate antibodies. IP$_3$ (Dojindo) was added at the beginning of the experiment when stipulated. Pull-down assay using recombinant proteins was performed similarly in TNE buffer with 10 µg GST fusion proteins mixed with recombinant IRBIT (1 µg) and/or recombinant Bcl2l10 (1 µg).

## Subcellular fractionation

Subcellular fractionation of HeLa cells was performed using Percoll gradient as described previously (*Williamson et al., 2015*). Briefly, HeLa cells were homogenized by passing through a 27 G 3/4" needle. Crude mitochondria were pelleted by centrifugation at 10,500 x $g$ for 10 min at 4°C and the resulting supernatant was further ultracentrifuged at 100,000 x $g$ for 1 hr at 4°C to isolate ER. Crude mitochondria were layered on top of a 30% Percoll gradient and ultracentrifuged at 95,000 x $g$ for 65 min at 4°C. Bands corresponding to MAMs and pure mitochondria were extracted from the gradient and diluted with PBS. MAM fraction was isolated by ultracentrifugation at 100,000 x $g$ for 45 min at 4°C and pure mitochondria by centrifugation at 6,300 x $g$ for 20 min at 4°C. The protein concentration of each fraction was determined using the Bradford assay (Bio-Rad) and equivalent amounts of protein (10 µg) were analyzed by immunoblotting with appropriate antibodies.

## Western blot

Proteins were separated by SDS-PAGE and transferred to a polyvinylidene difluoride (PVDF) membrane. The membrane was blocked for 1 hr at room temperature with 5% milk in PBS containing 0.05% Tween-20 (PBS-T) and immunoblotted with primary antibodies diluted in PBS-T + 3% milk for 1 hr at room temperature or 16 hr at 4°C. After washing with PBS-T, the membranes were incubated with an appropriate HRP-conjugated secondary antibody. Immunoreactive bands were detected with ECL Select Western Blotting Detection Reagents (GE Healthcare) or Immobilon Western Detection Reagents (Millipore) and captured using a luminescent image analyzer (LAS-4000 mini, GE healthcare).

## Western blot quantification and analysis

Band intensities were quantified using Fiji software (RRID:SCR_002285). For pulldown quantification, the band intensity of the protein of interest in the pulldown was normalized by the intensity of the GST band in the same lane. This ratio ($R_{GST}$) for a given condition was then normalized by the $R_{GST}$ of the control to obtain the relative pulldown value. Statistical significance was performed using $R_{GST}$ values.

For cleaved-PARP quantification, band intensity in cleaved-PARP Western blot (WB) was normalized by the intensity of the Actin band WB in the same lane. This ratio ($R_{Actin}$) for a given condition was then normalized by the $R_{Actin}$ of the control (DMSO) to obtain the relative cleaved-PARP value. Statistical significance was performed using $R_{Actin}$ values.

For phosphorylation quantification, band intensity in phosho-IRBIT or IRBIT WB was normalized by the intensity of the Actin band WB in the same lane. This ratio ($R_{Actin}$) for phosho-IRBIT at a time point was then normalized by the $R_{Actin}$ for IRBIT at the same time point to obtain the relative phosphorylation value. Statistical significance was performed using $R_{Actin}$ values.

For Bcl2l10 and IRBIT quantification in subcellular fractionation, band intensity in Bcl2l10 or IRBIT WB was normalized by the intensity of the loading control band in the same lane (GAPDH for Cytosol, IP$_3$R for MAM and ER). This ratio ($R_{loading}$) for a given condition was then normalized by the $R_{loading}$ of the control (DMSO) to obtain the relative band intensity. Statistical significance was performed using $R_{loading}$ values.

## 2D BN-SDS PAGE

Two dimension electrophoresis was performed as described previously (*Wittig et al., 2006*). Briefly, 200 µg of crude mitochondria were resuspended in solubilization buffer (50 mM NaCl, 50 mM imidazole pH 7, 2 mM 6-Aminohexanoic acid, 1 mM EDTA) were solubilized with digitonin (3 g/g of protein) for 10 min on ice and then centrifuged at 20,000 x $g$ for 20 min at 4°C. Glycerol and Coomassie blue G-250 were added to the supernatant and the mixture was loaded on a 6% acrylamide tricine gel. After running the gel at 4°C, the band corresponding to the loaded sample was excised and then incubated with 2x SDS-PAGE sample buffer and heated. After 20 min incubation in SDS-PAGE sample buffer at room temperature, the band was put on a 10% acrylamide Bis-Tris SDS gel and subsequent electrophoresis and immunoblot were performed as described in the Western blot section.

## Ca$^{2+}$ imaging

For cytosolic Ca$^{2+}$ measurement, MEF cells plated in 35 mm glass base dishes (Iwaki) were co-transfected with pEGFPC1 and either empty pBI-CMEF, pBI-CMEF-FLAG-Bcl2l10, pBI-CMEF-FLAG-IRBIT, pBI-CMEF-FLAG-IRBITS68A, pBI-CMEF-FLAG-IRBIT-FLAG-Bcl2l10 or pBI-CMEF-FLAG-IRBITS68A-FLAG-Bcl2l10 (1:3 ratio). For siRNA, HeLa cells plated in 35 mm glass base dishes were co-transfected with 0.5 µg of pEGFPC1 and 50 pmol of siRNA. 24 hr after plasmid transfection or 48 hr after siRNA transfection, cells were loaded with 5 µM Fura-2 AM (DOJINDO) for 30 min, then placed on the stage of an inverted microscope (IX-70; Olympus, Japan) and perfused with balanced salt solution (BSS, 20 mM Hepes, pH 7.4, 115 mM NaCl, 5.4 mM KCl, 1 mM MgCl$_2$, 10 mM glucose, and 2 mM CaCl$_2$). Cells were stimulated with ATP to induce transient intracellular Ca$^{2+}$ release or with thapsigargin to induce ER emptying. GFP-positive cells were identified before Ca$^{2+}$ imaging and variation of the ratio (R) of 340 nm/380 nm excited Fura-2 fluorescence of these cells was analyzed.

For cytosolic Ca$^{2+}$ measurement following drugs treatment, HeLa cells were incubated with 1 µM STS and 5 µM Fura-2 in BSS for 30 min before imaging or for 3h30 with DMSO (1/1000) or 20 µM TUN in culture medium followed by 30 min with 5 µM Fura-2 plus DMSO (1/1000) or 20 µM TUN in BSS before imaging. Ca$^{2+}$ imaging was performed as for cytosolic Ca$^{2+}$ measurement and the variation of the ratio (R) of 340 nm/380 nm was analyzed for 100 cells chosen randomly in the field.

For mitochondrial Ca$^{2+}$ measurement, MEF cells plated in 35 mm glass base dishes (Iwaki) were loaded with 2.5 µM Rhod-2 AM (DOJINDO) for 1 hr. Ca$^{2+}$ imaging was performed as for cytosolic Ca$^{2+}$ measurement, except that fluorescence was recorded with excitation at 550 nm and emission at 590 nm.

For mitochondrial Ca$^{2+}$ measurement after drug treatment, HeLa cells were loaded with 2.5 µM Rhod-2 AM (DOJINDO) for 1 hr. They were then incubated either with DMSO (1/1000) or 20 µM TUN in culture medium for 8 hr or in culture medium for 6h30 followed by 1h30 with 1 µM STS in culture medium. Images were then acquired with excitation at 550 nm and emission at 590 nm with the same exposure time for every condition. The fluorescence intensity of each of the cells in the field were then measured using Fiji software (RRID:SCR_002285). The fluorescence value of each cell was normalized by the mean fluorescence of the control condition (DMSO) giving the relative fluorescence of each cell. This relative fluorescence was used to calculate the mean relative fluorescence, the SEM and to analyze statistical significance.

## Apoptosis measurement

For active Caspase-3 staining, HeLa cells were cultured in 35 mm glass base dishes (Iwaki) and, when indicated, transfected with either empty pBI-CMEF, pBI-CMEF-FLAG-Bcl2l10, pBI-CMEF-FLAG-IRBITS68A or pBI-CMEF-FLAG-IRBITS68A-FLAG-Bcl2l10. 24 hr after transfection, cells were treated with DMSO (1/1000, Sigma-Aldrich) for 24 hr, 1 µM staurosporine (LKT Laboratories) for 4 hr, 2 µM thapsigargin (Calbiochem) for 24 hr or 20 µM tunicamycin (Sigma-Aldrich) for 24 hr and then stained using the Image-iT LIVE Green Caspase-3 and −7 Detection Kit (ThermoFisher Scientific) according to the manufacturer's instructions. Images were acquired with a fluorescence microscope (Biozero BZ-8100, Keyence).

For western blot analysis of apoptosis, cells were transfected, when indicated, with empty pHM6 or pHM6-IRBIT and treated, 24 hr after transfection, with DMSO (1/1000, Sigma-Aldrich) for 24 hr, 1 µM staurosporine (LKT Laboratories) for 4 hr, 2 µM thapsigargin (Calbiochem) for 24 hr or 20 µM tunicamycin (Sigma-Aldrich) for 24 hr. After treatment, cells were lysed in TNE buffer, the protein concentration of each sample was determined using the Bradford assay (Bio-Rad) and equivalent amounts of protein (10 µg) were analyzed by immunoblotting with appropriate antibodies.

## Immunofluorescence

Cells cultured on coverslips were transfected with pcDNA3.1-KDEL-GFP and 24 hr after transfection were fixed with 4% paraformaldehyde in PBS for 20 min at 37°C, then washed three times with PBS. Cells were then permeabilized and blocked for 20 min at room temperature with blocking buffer (PBS, 5% normal goat serum, 0.1% Triton X-100) before incubation with anti-Tom20 antibody (1/2000 dilution in blocking buffer) for 1 hr at room temperature. After three 5 min washes with PBS-T (PBS, 0.1% Triton X-100), cells were incubated with secondary antibody (goat anti-rabbit Alexa Fluor 568, Thermofisher Scientific) for 1 hr at room temperature and then washed for 5 min with PBS-T

three times. Coverslips were mounted with Vectashield (Vector Laboratories) and observed under a confocal fluorescence microscope (FV1000, Olympus) with a ×60 objective. Fluorescence images were analyzed by FV10-ASW software (Olympus). Colocalization coefficients were calculated with the Coloc 2 plugin of ImageJ software.

## Electronic microscopy

HeLa cells cultured on coverslips were fixed at room temperature for 3 hr in 2.5% glutaraldehyde in 0.1 M phosphate buffer pH 7.3, and then washed 3 times with 0.2 M sucrose in 0.1 M phosphate buffer pH 7.3. Specimens were then post-fixed with 1% $OsO_4$ in 0.1 M phosphate buffer pH 7.3 for 1 hr and stained en bloc with uranyl acetate. The specimens were then dehydrated through a graded series of ethanol and embedded in Epon 812. Ultrathin sections were examined with a transmission electronic microscope (Hitachi H-7100) after double staining with uranyl acetate and lead citrate.

## Sequence alignment

Nrz and Bcl2l10 sequences were found in the National Center for Biotechnology Information protein database. Sequence alignment was performed with the Clustal Omega tool at http://www.ebi.ac.uk/Tools/msa/clustalo/. An image of the alignment was obtained with Jalview software (RRID:SCR_006459) (*Waterhouse et al., 2009*).

## Statistical analysis

Statistical significance was analyzed using the Student's *t* test. Values in graphs were expressed as mean ± SEM, *p<0.05, **p<0.01, ***p<0.001.

## Acknowledgements

We thank Pr. Germain Gillet for providing pSG5-FLAG-Bcl2l10 vector. This work was supported by RIKEN (Special Postdoctoral Researcher [SPDR]), the Japan Society for the Promotion of Science (International Research Fellow) and JST International Cooperative Research Project–Solution Oriented Research for Science and Technology.

## Additional information

### Funding

| Funder | Author |
| --- | --- |
| Japan Society for the Promotion of Science | Benjamin Bonneau |
| RIKEN | Benjamin Bonneau |

The funders had no role in study design, data collection and interpretation, or the decision to submit the work for publication.

### Author contributions

BB, Conception and design, Acquisition of data, Analysis and interpretation of data, Drafting or revising the article; HA, KK, Acquisition of data, Analysis and interpretation of data, Drafting or revising the article; MH, Acquisition of data, Contributed unpublished essential data or reagents; HT-I, Acquisition of data, Analysis and interpretation of data; KM, Conception and design, Drafting or revising the article

### Author ORCIDs

Benjamin Bonneau, http://orcid.org/0000-0003-2302-5369
Katsuhiko Mikoshiba, http://orcid.org/0000-0002-3487-6970

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
