## [Decision Letter]

Thank you for submitting your article "IRBIT controls apoptosis by interacting with the BCl-2 homolog, Bcl2l10, and by promoting ER-mitochondria contact" for consideration by *eLife*. Your article has been favorably evaluated by Tony Hunter as the Senior Editor and three reviewers, including Richard S Lewis (Reviewer #1) - who is a member of our Board of Reviewing Editors – and David Yule (Reviewer #2).

The reviewers have discussed the reviews with one another and the Reviewing Editor has drafted this decision to help you prepare a revised submission.

Summary:

In this paper, Bonneau et al. describe a new role for IRBIT in promoting apoptosis through its interactions with a Bcl2 homolog, Bcl2l10, and the IP_3_ receptor. They present evidence that the three proteins form a complex at contact points between the ER and mitochondria, which inhibits IP_3_R-mediated Ca^2+^ release more than either Bcl2l10 or IRBIT can do alone. Further results support a proposed model in which apoptotic stimuli lead to dephosphorylation of IRBIT, releasing it and Bcl2l10 from the IP_3_R, and thereby increasing Ca^2+^ transfer from ER to mitochondria and promoting apoptosis. The paper is significant in presenting a mechanism underlying a new role for IRBIT in promoting apoptosis.

All three reviewers agreed that the paper is of significant interest, the paper is clearly written, and the data are mostly of high quality. However, several of the more important mechanistic conclusions need stronger experimental support. We would be willing to consider a revised version provided it contains new data and analysis that address the following major issues.

Essential revisions:

1) Experiments are needed to support the conclusion that dephosphorylated IRBIT controls the effect of endogenous Bcl2l10 on ER Ca^2+^ release and apoptosis. In Figure 5, overexpression of S68A IRBIT alone fails to affect Ca^2+^ release or apoptosis in HeLa cells, which is not consistent with the proposed role of dephosphorylated IRBIT in removing endogenous Bcl2l10 from the IP_3_R. These results raise the possibility that IRBIT only alter the effect of Bcl2l10 when the latter is overexpressed. New data are needed to show that endogenous Bcl2l10 itself affects Ca^2+^ transfer to mitochondria, using Bcl2l10 knockout or knockdown cells. In addition, the lack of effect seen with IRBIT S68A expression on Ca^2+^ release needs to be explained; perhaps stimulation with lower [ATP] will help reveal an effect.

2) The model in which IRBIT promotes apoptosis by removing Bcl2l10 from the IP_3_R and increasing Ca^2+^ transfer from ER to mitochondria cannot easily explain the effects of IRBIT on Tg-mediated apoptosis (Figure 4, Figure 5). Tg depletes Ca^2+^ from the ER, preventing prolonged Ca^2+^ transfer from ER to mitochondria; therefore, Tg would not be expected to invoke the IRBIT/Bcl2l10/IP_3_R mechanism or be sensitive to ER-mitochondrial contacts. The ability of IRBIT KO to have such a large effect on Tg-induced apoptosis (Figure 4) suggests critical targets other than the IP_3_R or induction of MAMs are involved. This issue needs to be addressed, as it raises the question of how much of IRBIT's effects on apoptosis are actually due to the Bcl2l10-IP_3_R pathway. To link this mechanism to apoptosis, evidence should be provided that the different stimuli that were used (especially STS and Tun) promote apoptosis specifically by increasing ER-mitochondrial Ca^2+^ transfer, and that their effects on IRBIT phosphorylation and localization of IRBIT/Bcl2l10 (Figure 4) correlate with apoptosis (Figure 5).

3) More data are needed to support a causative rather than correlative link between IRBIT and apoptosis progression. It is important to show that (1) apopotic stimuli enhance ATP-evoked ER-mitochondrial Ca transfer, and that IRBIT KO reduces this effect of apoptotic stimuli; and (2) release of IRBIT/Bcl2l10 is causally connected to enhanced apoptosis. #2 may be achieved using a mutant IRBIT that cannot be dephosphorylated (e.g., a phosphomimetic mutant), which would be predicted to stay associated with IP_3_R and reduce the effect of apoptotic stimuli on Ca^2+^ transfer to mitochondria.

4) To rigorously justify many of the important conclusions of the paper, changes in protein abundance or location as inferred from western blots need to be quantified. Band densities, number of repetitions, and statistical analyses should be shown. This applies to Figure 2, Figure 3, Figure 4, and 5A-B. These changes should be relatively straightforward, even allowing for repeats of some experiments.

---

## [Author Response]

*Essential revisions:*

*1) Experiments are needed to support the conclusion that dephosphorylated IRBIT controls the effect of endogenous Bcl2l10 on ER Ca^2+^ release and apoptosis. In Figure 5, overexpression of S68A IRBIT alone fails to affect Ca^2+^ release or apoptosis in HeLa cells, which is not consistent with the proposed role of dephosphorylated IRBIT in removing endogenous Bcl2l10 from the* IP_3_R*. These results raise the possibility that IRBIT only alter the effect of Bcl2l10 when the latter is overexpressed. New data are needed to show that endogenous Bcl2l10 itself affects Ca2+ transfer to mitochondria, using Bcl2l10 knockout or knockdown cells. In addition, the lack of effect seen with IRBIT S68A expression on Ca2+ release needs to be explained; perhaps stimulation with lower [ATP] will help reveal an effect.*

As proposed by the reviewers, the effect of IRBIT S68A on Ca^2+^ release may be masked due to high Ca^2+^ release in our experimental conditions. Ca^2+^ release experiments were carried out in MEF IRBIT KO cells, which indeed exhibit a strong Ca^2+^ release following ATP treatment. We then investigated the effect of IRBIT S68A in HeLa IRBIT KO cells, which show weaker Ca^2+^ release. Using these cells, we found that overexpression of IRBIT S68A alone increases IICR to the same extent as Bcl2l10 knockdown using siRNA. We believe that these data support the conclusion that dephosphorylated IRBIT controls the effect of endogenous Bcl2l10 on ER Ca^2+^ release. These data have been added to Figure 6 (Figure 6) and text related to these panels has been added to the second paragraph of the subsection “Unphosphorylated IRBIT inhibits Bcl2l10 function at the ER”.

Regarding the lack of effect of IRBIT S68A on apoptosis, these experiments have been carried out in HeLa cells containing endogenous IRBIT. It was previously proposed that the endogenous level of phosphorylated IRBIT may be high enough to saturate the interaction with IP_3_R as overexpression of IRBIT in cells containing endogenous IRBIT has no effect on Ca^2+^ release (see Ando et al., Mol Cell, 2006). So, in a similar manner, it is possible that the pool of endogenous IRBIT, which is dephosphorylated during apoptosis, is sufficient to remove the large majority of Bcl2l10 from ER membranes and then favors apoptosis. Overexpressing IRBIT S68A in these conditions may not have a significant effect on endogenous Bcl2l10 removal and then on apoptosis.

*2) The model in which IRBIT promotes apoptosis by removing Bcl2l10 from the* IP_3_R *and increasing Ca^2+^ transfer from ER to mitochondria cannot easily explain the effects of IRBIT on Tg-mediated apoptosis (Figure 4, Figure 5). Tg depletes Ca^2+^ from the ER, preventing prolonged Ca^2+^ transfer from ER to mitochondria; therefore, Tg would not be expected to invoke the IRBIT/Bcl2l10/*IP_3_R *mechanism or be sensitive to ER-mitochondrial contacts. The ability of IRBIT KO to have such a large effect on Tg-induced apoptosis (Figure 4) suggests critical targets other than the* IP_3_R *or induction of MAMs are involved. This issue needs to be addressed, as it raises the question of how much of IRBIT's effects on apoptosis are actually due to the Bcl2l10-*IP_3_R *pathway. To link this mechanism to apoptosis, evidence should be provided that the different stimuli that were used (especially STS and Tun) promote apoptosis specifically by increasing ER-mitochondrial Ca^2+^ transfer, and that their effects on IRBIT phosphorylation and localization of IRBIT/Bcl2l10 (Figure 4) correlate with apoptosis (Figure 5).*

Reviewers raised here a very important point. We agree that Tg depletes Ca^2+^ from the ER and then we can consider that Tg-dependent apoptosis does not rely on ER-mitochondria Ca^2+^ transfer.

However, it has been shown that inhibiting IP_3_R attenuates Tg-dependent apoptosis whereas IP_3_R stimulation accelerates Tg-dependent apoptosis (Luciani et al., Diabetes 2009 Feb; 58(2): 422-432). Thus, IP_3_R seems to play a role in cell death following Tg treatment a role and releasing Bcl2l10 from IP_3_R may then facilitate it. However, we agree that ER-mitochondria Ca^2+^transfer is probably not the most important trigger of Tg-dependent apoptosis.

On the other hand, Bcl2l10 was reported to interact with Bik and Bax (Rautureau et al., Cell Death Dis. 2012 Dec 13;3:e443.). Bik is a BH3-only protein which is localized at the ER thanks to a TM domain (Germain et al., 2002 May 17;277(20):18053-60). Bax can also localize to the ER where it induces apoptosis (Nutt et al., J Biol Chem. 2002 Jun 7;277(23):20301-8; Zong et al., J Cell Biol. 2003 Jul 7;162(1):59-69). Of note, Bik is involved in Bax recruitment to the ER (Mathai et al., J Biol Chem. 2005 Jun 24;280(25):23829-36). Interestingly, Bik is required for Tg-induced cell death (Lopez et al., Cell Death Differ. 2012 Sep;19(9):1459-69) and Tg promotes Bax dimerization at the ER which induces subsequent apoptosis (Zong et al., J Cell Biol. 2003 Jul 7;162(1):59-69).

Then after Tg treatment IRBIT dephosphorylation and subsequent removal of Bcl2l10 from ER membranes may promote Bik and Bax activity at the ER as Bcl2l10 is no longer present to interact with and inhibit these pro-apoptotic proteins.

This mechanism could explain the effect of IRBIT KO on Tg-induced cell death and the effect of Tg on IRBIT phosphorylation and IRBIT/Bcl2l10 localization. However, as we have not yet experimental proofs to support this model and for clarity reasons we decided to remove all data related to Thapsigargin from the manuscript and to focus on STS and Tunicamycin. Then as requested, we checked that these stimuli have a similar effect on IRBIT phosphorylation and localization of IRBIT/Bcl2l10. Thus Figure 4 has been changed accordingly. Moreover, we correlated this with apoptosis in Figure 5 (now Figure 6). Finally, we analyzed mitochondrial Ca^2+^ with Rhod-2 after STS and TUN treatment and we found that these stimuli significantly increase mitochondrial [Ca^2+^] whereas this effect is greatly attenuated in IRBIT KO cells (Figure 5). All together these new results show that STS and TUN acts on IRBIT and Bcl2l10 and promotes Ca^2+^ transfer to mitochondria.

*3) More data are needed to support a causative rather than correlative link between IRBIT and apoptosis progression. It is important to show that (1) apopotic stimuli enhance ATP-evoked ER-mitochondrial Ca transfer, and that IRBIT KO reduces this effect of apoptotic stimuli; and (2) release of IRBIT/Bcl2l10 is causally connected to enhanced apoptosis. #2 may be achieved using a mutant IRBIT that cannot be dephosphorylated (e.g., a phosphomimetic mutant), which would be predicted to stay associated with* IP_3_R *and reduce the effect of apoptotic stimuli on Ca^2+^ transfer to mitochondria.*

1) Experiments showing that STS and TUN treatment enhance ATP-induced Ca^2+^ release in WT cells have been performed (Figure 5). We also show in these figures that this effect of STS and TUN is significantly reduces in IRBIT KO cells. Moreover, in Figure 5, we no show that IRBIT KO also greatly decreases elevation of mitochondrial [Ca^2+^] consecutive to STS and TUN treatment. These new data indicate that IRBIT play a key role in apoptosis progression by facilitating ER-mitochondria Ca^2+^ transfer.

2) We generated several phosphomimetic mutants (mutations of Ser to Glu or Asp) of IRBIT to address this question. However, all the mutants generated failed to bind to the IP_3_BD of IP_3_R (see Figure 9). Phosphorylation of IRBIT required for the binding with IP_3_R occurs in an intrinsically disordered region (Ando et al., PLoS One. 2015 Oct 28;10(10):e0141569) and sequential phosphorylation of IRBIT may then be important to structure this region. Adding negative charge to mimic phosphorylation may then fail to reproduce the effect of phosphorylation. Then despite our effort, we were not able to conduct experiments which can answer this question.

Author response image 1.Western blot of GST-pulldown performed with GST-IP_3_BD on lysates of HeLa cells expressing FLAG-IRBIT (WT) or FLAG-tagged mutants of IRBIT.Mutations: SDDD: S71D, S74D and S77D; ADDD: S68A, S71D, S74D and S77D; DDDD: S68D, S71D, S74D and S77D; EDDD: S68E, S71D, S74D and S77D.**DOI:**
http://dx.doi.org/10.7554/eLife.19896.013

*4) To rigorously justify many of the important conclusions of the paper, changes in protein abundance or location as inferred from western blots need to be quantified. Band densities, number of repetitions, and statistical analyses should be shown. This applies to Figure 2, Figure 3, Figure 4, and 5A-B. These changes should be relatively straightforward, even allowing for repeats of some experiments.*

Quantification of Western blot have been performed. Relative band densities as well as statistical analysis have been added to Figure 2, Figure 4 and Figure 5 (Now Figure 6). Number of repetitions are now shown in figures legends. Moreover, the method used to quantify and analyze these western blots is now described in the Materials and methods section. Concerning Figure 3, we don’t understand what needs to be quantified as this figure just describes the subcellular localization of IRBIT and Bcl2l10. The text related to this figure in the initial manuscript was probably confusing and it has been changed to be clearer.